# Changing Sea Level, Changing Shorelines: Integration of Remote Sensing Observations at the Terschelling Barrier Island

Bene Aschenneller[1], Roelof Rietbroek[1], and Daphne van der Wal[1,2]

[1]Faculty of Geo-Information Science and Earth Observation (ITC), University of Twente, PO Box 217, 7500 AE Enschede, the Netherlands
[2]NIOZ Royal Netherlands Institute for Sea Research, Department of Estuarine and Delta systems, PO Box 140, 4400 AC Yerseke, the Netherlands

**Correspondence:** Bene Aschenneller (s.aschenneller@utwente.nl)

**Abstract.** Sea level rise is associated with increased coastal erosion and inundation. However, the effects of sea level change on the shoreline can be enhanced or counteracted by vertical land motion and morphological processes. Therefore, knowledge about the individual contributions of sea level change, vertical land motion and morphodynamics on shoreline changes is necessary to make informed choices for climate change adaptation, such as applying coastal defence measures. Here, we assess the potential of remote sensing techniques to detect a geometrical relationship between sea level rise and shoreline retreat for a case study at the Terschelling barrier island at the Northern Dutch coast. First, we find that sea level observations from satellite radar altimetry retracked with ALES can represent sea level variations between 2002 and 2022 at the shoreline when the region to extract altimetry timeseries is chosen carefully. Second, results for cross-shore timeseries of satellite-derived shorelines extracted from optical remote sensing images can change considerably dependent on choices made for tidal correction and parameter settings during the computation of timeseries. While absolute shoreline positions can differ on average by more than 200 m, the average trend differences are below $1 \, \mathrm{m \, yr^{-1}}$. Third, by intersecting the 1992 land elevation with time variable sea level, we find that inundation through sea level rise caused on average $-0.3 \, \mathrm{m \, yr^{-1}}$ of shoreline retreat between 1992 and 2022. The actual shoreline movement in this period was on average between $-2.8 \, \mathrm{m \, yr^{-1}}$ and $-3.2 \, \mathrm{m \, yr^{-1}}$, leading to the interpretation that the larger part of shoreline changes at Terschelling is driven by morphodynamics. We conclude that the combination of sea level from radar altimetry, satellite derived shorelines and land elevation provides valuable information about the influence of sea level rise, vertical land motion and morphodynamics on shoreline movements.

## 1 Introduction

At the end of this century, global mean sea level is expected to rise by 0.56 m (middle-of-the-road scenario SSP2-RCP4.5) compared to the period 1986–2005 as computed in an ensemble-median projection of CMIP6 models (Hermans et al., 2021). In the Wadden Sea, projected sea level rise is close to global sea level rise with values between 0.41 m ± 0.25 m (RCP2.6) and 0.76 m ± 0.36 m (RCP8.5), derived from CMIP5 for the period 2018–2100 given as the 5–95% confidence interval (Vermeersen et al., 2018). Apart from saltwater intrusion, sea level rise also poses a higher risk of coastal hazards like floodings and storm surges, and has the potential to erode and inundate coastal areas. With an estimated global population of 267 million

people living in coastal areas below 2 m above mean sea level (Hooijer and Vernimmen, 2021), sea level rise is expected to lead to immense damage and costs for humanity and ecosystems (Hinkel et al., 2014; Schuerch et al., 2018). In order to enable timely and appropriate implementation of adaptation and defence strategies, it is therefore important to provide coastal zone managers and other coastal stakeholders with accurate information about current and expected sea level rise and its effect on shoreline movements.

Shoreline movements are not only caused by changes in sea level, but also by vertical land motion and morphological changes. Examples for morphological changes are sediment transports by currents, waves and wind, or sediment accumulation by vegetation. While sea level and vertical land motion are relatively well covered by observations from radar altimetry, tide gauges and GNSS (Global Navigation Satellite System), the complex morphodynamic contributions to shoreline changes and their feedback to rising sea levels are harder to quantify. More than five decades ago, Bruun (1962) first suggested a model where with rising sea level sediment is displaced seaward along a profile, the so called Bruun Rule. Although the Bruun Rule is often criticised for using unlikely assumptions and missing the full complexity of morphological changes (e.g. Cooper and Pilkey, 2004), it still plays a substantial role in today's research of the link between sea level rise and shoreline retreat (e.g. Vousdoukas et al., 2020; D'Anna et al., 2021; Atkinson et al., 2018).

Nowadays, there are several decades of remote sensing data available for coastal monitoring (Laignel et al., 2023). Instead of using the Bruun Rule, we suggest an alternative approach using observations for sea level and vertical land motion in combination with estimates of shoreline changes to quantify the geometrical relation between sea level and shoreline changes. In contrast to the limited number of previous observational studies (Le Cozannet et al., 2014), we aim at developing a method based on remote sensing datasets that can potentially be applied globally in the future. In this paper, we assess the potential of remote sensing estimates for sea level and for shoreline changes by comparing them individually to complementary data sets covering the same processes, and by combining them in order to study their interplay in terms of geometrical changes over a maximum time period of 30 years (1992–2022).

## 1.1 Sea surface heights

Sea level variations are observed with two techniques, locally with tide gauges and globally with satellite radar altimetry. Radar altimetry captures absolute sea level changes, the combined effect of mass- and volume changes relative to a reference ellipsoid. In contrast, tide gauges register the relative sea surface heights from a station fixed to the ground, therefore these observations are also influenced by vertical land motion, including land subsidence and uplift. Vertical land motions can themselves lead to significant relative sea level changes (e.g. Pfeffer and Allemand, 2016; Santamaría-Gómez et al., 2012) and can be determined either directly using for example geodetic GNSS measurements, or indirectly, from the difference between relative sea surface heights from the tide gauges and absolute sea surface heights from satellite altimetry (e.g. Wöppelmann and Marcos, 2016; Kleinherenbrink et al., 2018; De Biasio et al., 2020).

Retrieving altimetric sea surface heights in coastal areas is especially challenging. First, reflections from land in the altimeter footprint result in distorted signals, which require the application of specialised retracking algorithms to extract parameters such as the sea surface height. Second, the common geophysical and path delay corrections are not always available in the required temporal and spatial resolution to capture the small-scale processes near the coast. Retracking algorithms and corrections for coastal applications of satellite radar altimetry are continuously improved; currently, sea surface heights as close as 1-5 km to the coast can be retrieved (e.g. Birol et al., 2021; Vignudelli et al., 2019).

Satellite altimetry and tide gauges both observe sea level variations, but on different spatial and temporal scales. Tide gauges are almost exclusively installed on coasts and therefore lack spatial coverage, but they provide measurements with a high temporal resolution down to a few minutes (e.g. Holgate et al., 2013). On the other hand, satellite radar altimetry provides global coverage with a lower temporal resolution (e.g. Morrow et al., 2018; Vignudelli et al., 2011). The typical measurement frequency is about two to four observations per month, depending on the satellite's period of revolution around the Earth and the total number of altimetric satellites currently in orbit.

Despite these differences, observations from radar altimetry and tide gauges have been successfully combined for example to study the nearshore sea surface height changes due to strong currents, winds or bathymetry. Cipollini et al. (2017) compared non-retracked altimetry and tide gauge records at the coast of the UK by selecting matching pairs of observations in a radius of 0-200 km according to the smallest root mean square difference (RMSD). They found average correlations between 0.45 and 0.75 and RMSD values between 3.8 cm and 5.8 cm. Birol et al. (2021) did a similar study in six large coastal regions around the world and found an average correlation value of 0.77 for varying distances ($\sim$ 160 - 300 km) to the respective tide gauges.

## 1.2 Shoreline positions

The terms "coastline", "shoreline" and "land-water interface" are not used uniformly in the existing literature. Inspired by e.g. Boak and Turner (2005), we use these terms here as follows. A coastline describes the stretch along the coast, including both land and water surfaces. The land-water interface is the dynamic boundary between land and water. The shoreline is a proxy for the ideal, instantaneous land-water interface. In this study, we use two different techniques to observe shoreline positions and their temporal evolution. These are the detection of shorelines from optical satellite images, and the derivation of shorelines by intersecting land elevation data with a plane at sea surface height. Both realisations of the shoreline position refer to the same morphological feature. Their comparability depends on the respective observation uncertainties, the careful handling of different reference systems and the application of tidal corrections.

Shoreline positions extracted from optical satellite images are in the preceding literature usually referred to as satellite-derived shorelines. When working with images from optical satellite missions, there is usually a trade-off between spatial resolution and revisit period. The group of sensors with moderate resolution (about 250 m - 1000 m pixel size), such as Terra/Aqua MODIS, Envisat MERIS or Sentinel3 OLCI, have high revisit periods (about 0.5 - 3 days), but images are usually too

coarse to extract shoreline geometries. The most commonly used optical sensors for shoreline extraction are high resolution sensors (about 5 m to 30 m pixel size). Since 1999, these satellites often carry additional panchromatic sensors that generate black and white images with a finer resolution, and can be used to downscale the multispectral images. Examples are the long-term Landsat missions (30 m resolution of multispectral channels, with a 15 m panchromatic band) with a revisit period of 16 days, Sentinel-2 MSI (10-20 m resolution) with a revisit period of 10 days (single satellite) or 5 days (two satellites in tandem) and long-term SPOT (5-20 m) with a revisit period of 26 days. Of these missions, SPOT is the only one whose data is not freely available. Finally, there are several commercial satellite missions with very high resolution (< 5 m) and short revisit periods (about 1-5 days) such as IKONOS, QuickBird, WorldView, or the cube satellite constellations by PlanetScope/Maxar. A more detailed review of optical satellite missions is given in Huang et al. (2018).

The process of extracting the shoreline from optical images starts usually by separating between land and water pixels. The easiest way to achieve this is to use a single band, which would preferably be one of the infrared bands where the differences in reflectance between water and land are the highest. The image of this band can be converted into a binary image by applying a threshold (e.g. Frazier and Page, 2000; Pardo-Pascual et al., 2012). This threshold can be chosen by a try-and-error procedure, or by computing it for example by using Otsu's method. Next to thresholding, the use of water indices (the ratios of differences between bands) is very common to separate between land and water surfaces. There are several indices in use, where the choice depends on the type of the surroundings. For example, the Modified Normalised Difference Water Index (MNDWI) (Xu, 2006) was developed with the intent to better distinguish between water and buildings than the Normalised Difference Water Index (NDWI) (McFeeters, 1996). Usually the computation of an index is followed by the application of a threshold (e.g. Luijendijk et al., 2018; Dai et al., 2019; Almeida et al., 2021; Palomar-Vázquez et al., 2023), possibly also in combination with a classification (e.g. Vos et al., 2019b). Finally, there are advanced procedures to extract the shoreline at sub-pixel resolution, for example by using a marching squares algorithm to derive the shoreline contour (e.g. Bishop-Taylor et al., 2019a; Vos et al., 2019b) or by modelling the gradient of reflectances with polynomials and extracting the coordinates with the maximum gradient (e.g. Pardo-Pascual et al., 2012; Almonacid-Caballer et al., 2016; Sánchez-García et al., 2020).

The second proxy for the shoreline position used in this study is the intersection of a digital elevation model (DEM) with a horizontal plane at sea surface height. The land elevation data are often gained with airborne laser altimetry (LiDAR) or photographs, and bathymetric observations. When the DEM is available as gridded data, this can be done by extracting the contour line at sea level (e.g. Parker, 2003). Often, land elevation data are distributed on profiles along cross-shore transects. The approach presented by Stockdon et al. (2002) uses linear regression to fit a function through the elevation data of a part of the profile, e.g. ± 0.5 m around the shoreline, and evaluates this function at sea level. The error of this shoreline position compared to estimates from GPS surveys was found to be between ± 1.1 m and ± 1.4 m, averaged over a stretch of 60 km (Stockdon et al., 2002). This approach is applied relatively often in the existing literature (e.g. Morton et al., 2005; Robertson et al., 2004), although usually for a limited number of LiDAR overflights. Do et al. (2019) extracted shorelines derived from the yearly JARKUS dataset at the Dutch coast over a period of 25 years. However, all applications so far focused on extracting

the shoreline at one tidal datum, for example at mean high water, without testing the influence of sea level changes on the shoreline position.

## 1.3 Study area

As a study area we chose the barrier island of Terschelling, that lies in a row of barrier islands separating the North Sea from the Wadden Sea at the Northern Dutch and German coast (Fig. 1a). We selected this study area because of its suitability for validating our method; it houses two tide gauges and a GNSS station, is covered by yearly LiDAR and bathymetry observations and its orientation is not parallel to the ground tracks of the satellite altimeters. This configuration allows us compare the respective local and remote-sensing observations, and to include the influence of vertical land motion. Additionally, Terschelling

has a sandy beach, the type of beach that most available tools to extract satellite-derived shorelines are tailored to.

    The island is approximately 28 km long and 4 km wide. The amount of longshore sediment transports is estimated to lie between 0.8 and $1.2 \cdot 10^6 \mathrm{m}^3/\mathrm{yr}$ in eastern direction (Ridderinkhof et al., 2016). In the West and in the East of Terschelling, the Vlie inlet and the Borndiep inlet connect the North Sea with the Wadden Sea. The coastal sections close to the inlets are

characterised by ebb tidal deltas with shoals, as well as a spit on the side of the Vlie inlet. We focus especially on the almost straight northern sandy beach where there are no groynes, harbours or other hard, artificial structures. There was however one shoreface nourishment carried out in 1993 (Brand et al., 2022).

    Short-term sea level variations at Terschelling are dominated by diurnal tides with a tidal range of 1.2 m–2.8 m during

neap tide and spring tide, respectively. The average wave height is 1.5 m with a mean period of 8 seconds coming from west to north-east direction. During storms, the wave heights can increase to 5–6 m, with an increased period of 10-15 seconds (Quataert et al., 2020). The long-term closure depth is reported to range between 4 m and 10 m, increasing from the west to the east of Terschelling (Marsh et al., 1999).

## 1.4 Linking shoreline change to sea level change

In this paper, we bring together observations of vertical sea level heights and horizontal shoreline positions to investigate the geometrical influence of past and future sea level changes on shorelines. Additionally, we will compare estimates that describe similar processes in order to illustrate the uncertainties in the underlying observations.

    After describing the datasets and the required post-processing steps in section 2, we start by evaluating the ability of offshore

altimetry observations to capture sea level variations at the coast by comparing altimetric sea level anomalies to sea surface heights from tide gauges (Sect. 3.1). The relative sea level from the tide gauges is corrected for vertical land motion using the GNSS observations. A first result is then an altimetry timeseries that best represents sea level at the coast. Furthermore, the northern sandy beach of Terschelling is covered by the JARKUS dataset providing yearly observations of topographic and bathymetric heights along transects. By intersecting these profiles with a plane at sea level height we get an estimate of the

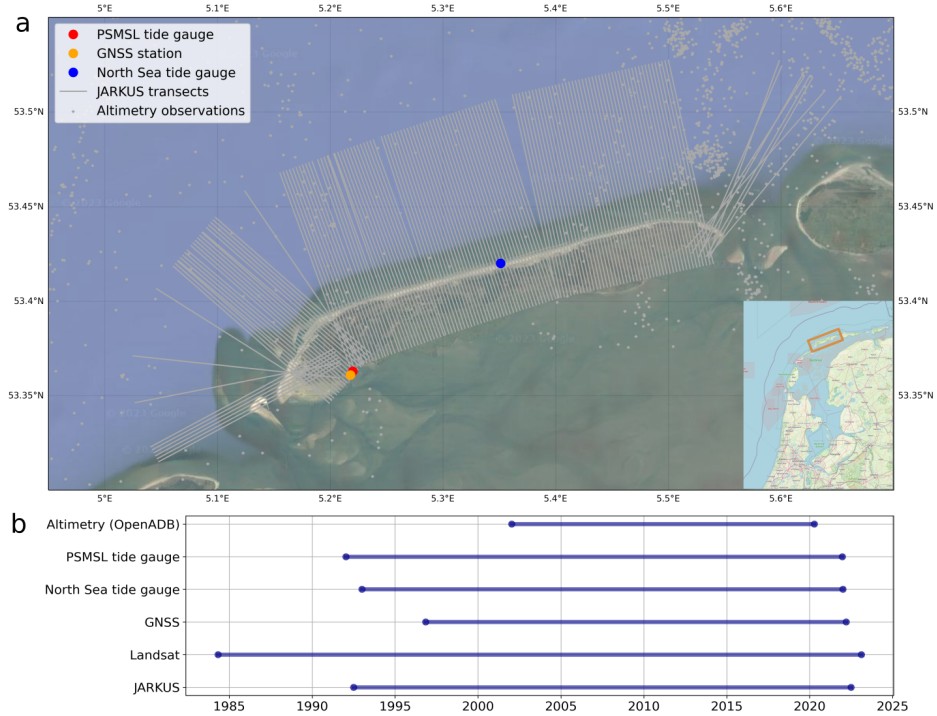

**Figure 1.** a: Locations of the PSMSL tide gauge coupled with the GNSS station, the North Sea tide gauge, the positions of the JARKUS transects and the closest of the offshore altimetry observations. Background image from Google Map tiles using cartopy.io.img_tiles (© Google Maps), inlay by OpenStreetMap (© OpenStreetMap contributors 2021. Distributed under the Open Data Commons Open Database License (ODbL) v1.0.). b: Time spans of the input data. For the two tide gauges and JARKUS we manually reduced the dataset to the period 1992–2022.

shoreline position (Sect. 3.2). A second estimate for the shoreline position comes from optical remote sensing images, here we use the software CASSIE by Almeida et al. (2021) with Landsat satellite images (Sect. 3.3 and 3.4). We compare both shoreline estimates to validate them and to evaluate the achievable accuracy (Sect. 3.5). In order to be able to detect a possible geometrical connection between sea level rise and shoreline changes, we aim at studying long time periods. From the chosen datasets, the longest overlapping period covered is 2002–2020, limited by the retracked altimetry data distributed by the Open Altimetry Database (OpenADB) (Fig. 1b). An overview over the entire workflow is given in Fig. 2.

## 2 Data

### 2.1 Sea level anomaly from coastal satellite altimetry

We downloaded altimetric sea surface heights retracked with ALES (Passaro et al., 2014) in 1 Hz sampling rate from the Open Altimetry Database (OpenADB) (DGFI, 2023). Additionally to the ALES retracking, DGFI (2023) subjected this dataset to

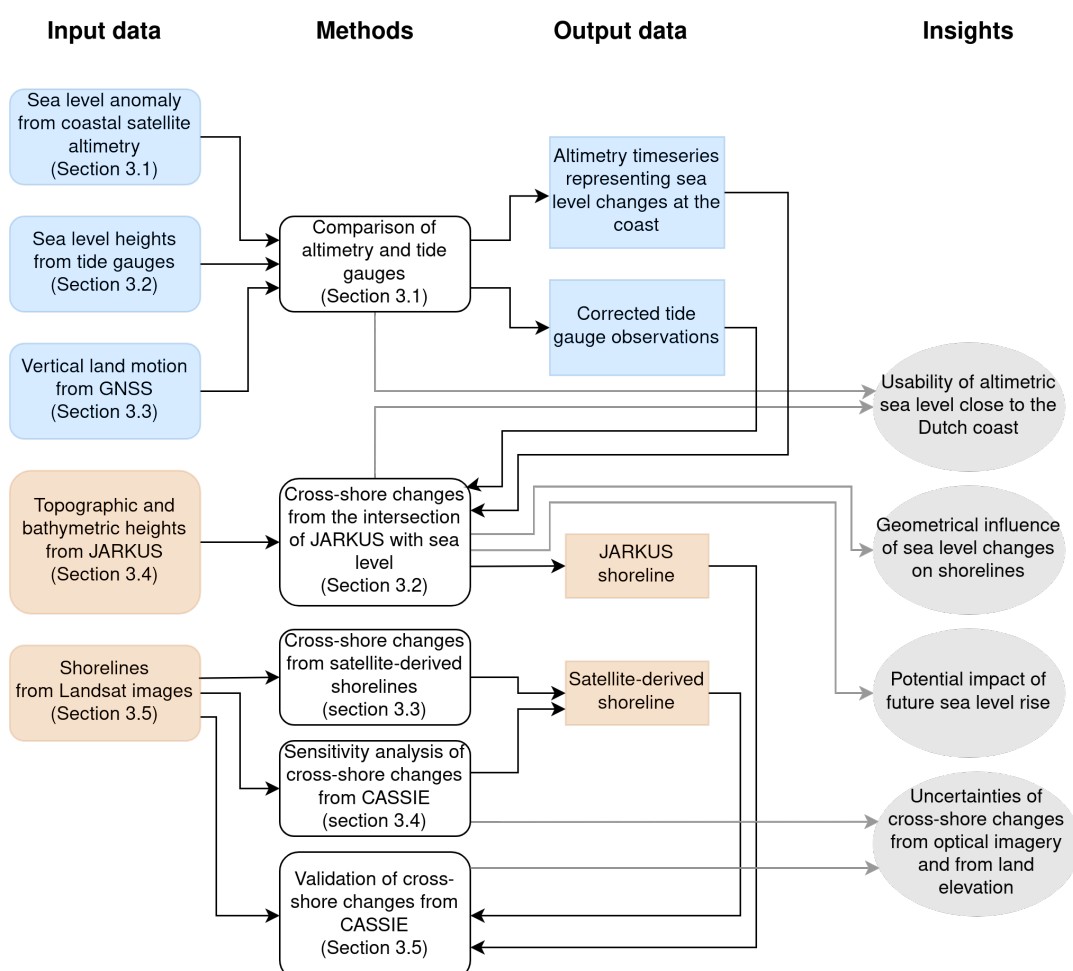

**Figure 2.** Workflow of this paper, with input and output data for sea level components in blue and for shoreline components in brown.

several pre-processing steps described in detail by (Oelsmann et al., 2021) and summarised in the following. First, the common corrections for path delay (wet and dry troposphere, ionosphere), as well as the geophysical corrections (atmospheric pressure, ocean tides, load tides, solid earth tides, pole tides, sea state bias) were applied. Details to the respective models used for the corrections can be found in (Oelsmann et al., 2021, Table 1). Second, sea surface heights from different satellite missions are cross-calibrated in a multi-mission crossover analysis. Third, the observations are interpolated on 1 Hz nominal tracks of the respective missions. Fourth, outliers were removed, defined as observations where the sea level anomaly is larger than 2 m, where the difference of the respective observation to the running median over 20 points along track is larger than 12 cm or where the difference of two consecutive points along track is larger than 8 cm.

We downloaded sea surface heights retrieved from Envisat, Jason-1, -2, -3, SARAL and Sentinel-3A/B, in total covering the period 21.06.2002 – 12.04.2020. The extracted data covers an area of about 200 x 300 km North of Terschelling. Additionally to the above mentioned outlier rejection already done by DGFI, we excluded data points as recommended on the OpenADB website according to the following rules, (1) distance to coast < 3 km, (2) |sea level anomaly| > 2.5 m, (3) significant wave height > 11 m and (4) fitting error on the normalised leading edge (ALES) > 0.20 m. In the following, we continue to work with sea level anomalies derived by subtracting the mean sea surface from the instantaneous sea surface heights. Mean sea surface data from DTU18 (Andersen et al., 2018) is provided together with the OpenADB data.

## 2.2 Sea level heights from tide gauges

We use tide gauge data to assess the potential of satellite altimetry to determine long term sea level variability close to the coast. At Terschelling, observations from two tide gauges are available that differ in location and temporal resolution. From both data sets, we extract the period from January 1992 to December 2021.

The North Sea tide gauge (station Terschelling Noordzee, Rijkswaterstaat (2022b)) provides water levels at 10-minute resolution. For the comparison with altimetry, tide gauge data needs to be corrected for the response of the sea surface to atmospheric pressure changes (also called "inverted barometer correction" or "IB correction"). We compute the IB correction using the following relationship from Ponte (2006):

$$\eta^{ib} = -\frac{P_a - \bar{P}_a}{\rho g}, \tag{1}$$

where $\eta^{ib}$ is the sea level change in response to the atmospheric pressure change, $P_a$ is the local sea level pressure, the overbar indicates the spatial average over the entire ocean surface, $\rho$ is the ocean density and $g$ the gravitational acceleration. We use monthly mean sea surface pressure from the ERA5 atmospheric reanalysis (Hersbach et al., 2022) for $P_a$ and the constants $\rho$ = 1027 $\text{kg/m}^3$ and $g$ = 9.81 $\text{m s}^{-2}$.

Furthermore, the tide gauge observations have to be corrected for vertical land motion. We use the vertical component of GNSS observations (see Sect. 2.3) and directly subtract them from the tide gauge timeseries. This approach is possible as we aim not at comparing absolute sea surface heights between altimetry and tide gauge, and focus instead on the temporal variability. Before subtracting, the GNSS observations are interpolated to each time step of the tide gauge records. As the time period 01.01.1992–03.11.1996 is not covered by GNSS observations, we extrapolate the trend and seasonal signal of the available GNSS observations back in time.

Additionally, the observations from the North Sea tide gauge in 10-minute resolution also have to be corrected for tides, for which we test several options:

1. The tidal models EOT20 (Hart-Davis et al., 2021) and FES2014 (Lyard et al., 2021), using the tidal prediction software from AVISO[1]. The ALES altimetric sea surface heights from OpenADB were tidally corrected with FES2014. However, tide models have issues in shallow regions close to the coast (e.g. Piccioni et al., 2018), therefore we test other tidal corrections as well.

2. Butterworth filter with order = 3 and cut-off frequency = 30 days. The butterworth filter is a low-pass filter that causes almost no spurious oscillations in the time domain even with a reasonable sharp cut-off in the frequency domain (Roberts and Roberts, 1978).

3. T_Tide, a software to perform a harmonic analysis of the tidal signal to empirically determine the tidal correction from the observations. T_Tide was developed by Pawlowicz et al. (2002) for matlab, here we use the python version[2] that currently can only handle timeseries shorter than the period of the nodal tidal cycle of 18.6 years. We therefore compute the tidal correction from the first 5 years of the timeseries (01.01.1992 - 31.12.1996) and apply this correction to the entire period. For the harmonic analysis, the shallow water constituents are taken into account.

A second set of tide gauge data is available from the Permanent Service for Mean Sea Level (PSMSL). PSMSL distributes monthly averaged RLR (Revised Local Reference) data from a tide gauge in the harbour of West-Terschelling (station 23, Holgate et al. (2013)), adjacent to the Wadden Sea. We correct the PSMSL dataset for atmospheric pressure changes and for vertical land motion as described above for the North Sea tide gauge data. As this data set is already monthly averaged, we assume that it does not contain any significant tidal signals.

## 2.3 Vertical land motion from GNSS

Observations of Vertical Land Motion (VLM), i.e., subsidence or uplift of the ground, are required to relate the relative sea level variations from the tide gauges to the absolute sea level variations from altimetry. We obtained observations of VLM from a permanent GNSS (Global Navigation Satellite System) station situated in the harbour of Terschelling (documented by Delft (2022)), close to the PSMSL tide gauge. This GNSS station provides weekly measurements starting on 03.11.1996 until today. We use the solution processed by the Nevada Geodetic Laboratory (NGL14) (Blewitt et al., 2018) and distributed by SONEL (SONEL, 2022) as this solution has the largest coverage over the entire operating period back to November 1996.

The GNSS time series contains discontinuities from antenna and receiver changes. The respective dates are documented on the station's website (Delft, 2022). We estimate the magnitude of the offsets by fitting a step function in a least squares adjustment to the timeseries, where each interval covers the period between two instrumental changes. Most offsets have a magnitude in the order of 1-2 mm which corresponds to the achievable accuracy of long term GNSS measurements. Therefore, not every estimated offset is necessarily significant, and not every instrumental change leads to a discontinuity in the time series (Fig. A1). Taking all offsets into account would lead to over segmentation and would eliminate the long term physical signal.

---

[1]https://github.com/CNES/aviso-fes
[2]https://github.com/moflaher/ttide_py

Determining which offsets are significant is a more or less subjective choice. Gazeaux et al. (2013) asked several groups to estimate time and magnitude of offsets in a simulated dataset. They found that from the variety of techniques applied, manual methods yielded overall better results than automated methods.

Here, we decide to manually remove one, two or three of the bigger offsets (with 9 mm, 4.5 mm and 3.8 mm respectively) in order to get a time series clean of artificial jumps but still containing the signal of VLM. The resulting VLM rates are summarised in Table 1, together with estimates from other publications for the same GNSS station. These estimates cover slightly different time periods, but when assuming that VLM rates are stable over approximately four years, we see a rather wide spread between -0.18 +/- 0.11 $\mathrm{mm\,yr}^{-1}$ (Gravelle et al., 2023, ULR7A) and $-0.63 \pm 0.43\,\mathrm{mm\,yr}^{-1}$ (Shirzaei et al., 2021). The differ-

ences in these outcomes of VLM rates indicate an uncertainty that approaches the magnitude of the signal. Another issue is that GNSS can only measure the component of VLM that takes place above the base of the GNSS station. Nevertheless, rates of GNSS height observations are currently the most accessible and up-to-date estimates of VLM. We therefore continue to work with the GNSS timeseries that results from removing the two largest offsets (version 2), as its VLM rate of -0.50 $\mathrm{mm\,yr}^{-1}$ fits best in the range of estimates from earlier publications.

There are several possible causes for this small rate of subsidence. Natural causes comprise glacial isostatic adjustment, however this trend was reported to be slightly positive over the Wadden Sea (Simon et al., 2018). Another reason could be sediment compaction, where only the layer between the base of the GNSS station and the ground leads to a vertical movement of $-4.1 \pm 1.8 \mathrm{mm\,yr}^{-1}$ at the Terschelling GNSS site (Karegar et al., 2020). Movement in this layer is not observed by direct

GNSS observations, but the overall layer susceptible to compaction continues about 16 m beneath the base of the GNSS station. Potential anthropogenic causes could be gas extraction from a small gas field in West-Terschelling and several gas fields in East-Ameland, as well as a salt extraction field on the other side of the Wadden Sea in Haveland (Fokker et al., 2018).

| Source | Period | VLM rate in $\mathrm{mm\,yr}^{-1}$ |
|---|---|---|
| This paper, original time series with all discontinuities | 11/1996–03/2022 | -1.15± 0.47 |
| This paper, version 1: Removing 9 mm-offset | 11/1996–03/2022 | - 0.69 ±0.43 |
| This paper, version 2: Additionally removing 4.5 mm-offset | 11/1996–03/2022 | -0.50 ± 0.42 |
| This paper, version 3: Additionally removing 3.8 mm-offset | 11/1996–03/2022 | -0.40 ± 0.42 |
| NGL14 (Blewitt et al., 2016) | 11/1996–03/2022 | -0.57 ± 0.41 |
| ULR7A (Gravelle et al., 2023) | 01/2000–12/2020 | -0.18 ± 0.11 |
| Shirzaei et al. (2021) | 10/1996–10/2019 | -0.63 ± 0.43 |

**Table 1.** Summary of rates of vertical land motion found from the NGL14 dataset when removing one, two or three of the biggest offsets, in comparison to the results of other publications over different periods. Error margins for results of this paper are given in the 1-99 % confidence interval.

## 2.4 Topographic and bathymetric heights from JARKUS

We use land elevation data to intersect the beach height profile with sea level as a proxy for the shoreline. The Dutch coast is covered by yearly observations of height profiles above and below water carrierd out by the Dutch Ministry of Infrastructure and Water Management (Rijkswaterstaat) since 1965. These measurements relative to NAP (Normaal Amsterdams Peil) are provided through the JARKUS dataset (JAaRlijkse KUStmeting, "Annual Coastal Measurement") (Rijkswaterstaat, 2022a; Pot, 2011; Minneboo, 1995). The data are provided along transects with a spacing of about 250 m–500 m (Minneboo, 1995; Athanasiou et al., 2019) and a spatial resolution in cross-shore direction of 5 m. Terschelling is covered by 217 transects (see Fig. 1). The orientation of the transects remains constant over time.

The height profiles are obtained from topographic and bathymetric measurements. The topography above the waterline was until 1996 observed with airborne stereo-photogrammetry (Pot, 2011), and since 1996 with airborne laser altimetry (LiDAR). The standard deviation of the LiDAR observations is estimated to lie between 10 and 15 cm (de Graaf et al., 2003). The bathymetry below the waterline is observed with single- and multi-beam echo sounding from ships (Pot, 2011; Wiegmann et al., 2002) to an extent where a depth of approximately 8 m below NAP is reached. Bathymetry observations are made during high tide, while topography observations are made during low tide, resulting in an overlap area.

## 2.5 Shorelines from optical satellite images

In order to assess the potential of satellite-derived shorelines for studying long-term shoreline changes, we use one of the available algorithms to extract shoreline positions from optical satellite images, CASSIE (Coastal Analyst System from Space Imagery Engine, Almeida et al. (2021)). As CASSIE runs entirely on Google Earth Engine, it is not required to download the images. The cloud computation makes CASSIE a good candidate for upscaling the methodology to a global application in the future.

After image pre-processing, the CASSIE software first computes the NDWI (Normalised Difference Water Index) and creates a histogram of NDWI values for each pixel and for each image. Second, the pixels per image are classified by computing the Otsu threshold from the histogram. By default, two classes (land, water) are used. For estuaries and other environments where the distinction between land and water is not clear, it is possible to use a multilevel Otsu thresholding with three classes (land, water, intertidal; water and intertidal are afterwards grouped together). Third, the features in the resulting binary image are converted to polygons. The first intersection of a polygon with the transect is defined as the shoreline. Finally, the so found shoreline contour is smoothed with a moving-average filter (Almeida et al., 2021).

When using CASSIE, data from multiple missions are available, i.e. surface reflectances from Landsat (5, 7, 8, 9) and top-of-atmosphere reflectances from Sentinel-2 (Almeida et al., 2021). Here we choose to use only Landsat images due to concerns about consistency when mixing surface and top-of-atmosphere reflectances. The analysis region is defined by an area of

interest that can be drawn or imported as a .kml-file. For the selection of images, we use the full available timespan 30.04.1984–22.02.2023 with a cloud cover of less than 50 %. Images where the beach is not visible due to clouds were manually removed from the selection. Next, the user has some control over the features detected as shorelines by defining a baseline and setting the spacing and extent of transects that are created automatically perpendicular to this baseline. This step is important, as CASSIE

detects only shorelines that intersect with a transect. We experimented using different baselines and extent parameters, as well as choosing 2 or 3 Otsu thresholding classes. Our experiments showed better shoreline detection when using the settings for 3 classes, as it reduced the number of transitions between bare sand and vegetation detected as shorelines.

Besides using CASSIE, we also extracted satellite-derived shorelines for five years between 2015 and 2020 from CoastSat

(Vos et al., 2019b). In CoastSat, the images are downloaded from Google Earth Engine and processed locally. The study area is therefore restricted by hardware storage size required to store the images, as well as by the recommendation to not request more than 100 $km^2$ at the same time to avoid slow processing. Additionally, we encountered a problem with the cloud masking that in the majority of images classified the beach as clouds and consequentially removed all information about shorelines. The cloud mask stems from the USGS quality assessment band computed with the CFMask algorithm. We tested switching

the parameters that control the type of cloud that is masked, but were not able to achieve satisfying results. In the end, we deactivated cloud masking and manually classified the images for these five years according to their subjective usability for shoreline detection.

## 3   Methodology

### 3.1   Comparison of altimetry and tide gauges

We compare the temporal variability of retracked offshore altimetry observations against the in-situ sea level measurements from the North Sea tide gauge and the PSMSL tide gauge. The goal is to find the altimetry timeseries that best represents the sea level changes at the shoreline over the 18-year period (January 2002–March 2020) available from OpenADB. The workflow how altimetry and tide gauge observations were made comparable in terms of corrections for tides, atmospheric pressure and vertical land motion is summarised in figure A2, and described in more detail in sections 2.1, 2.2 and 2.3.

We build altimetry timeseries by dividing the area into 25 x 25 $km$ cells and binning the OpenADB ALES retracked sea level anomalies into these cells. From each of the resulting timeseries, we remove outliers whose values exceed three times the standard deviation ($3\sigma$ rule). Next, we compute monthly averages of altimetry observations weighted with the inverse of the distance per observation to the cell centre.

The monthly averaged altimetry timeseries per cell are then compared to both tide gauges, the monthly PSMSL tide gauge and the North Sea tide gauge. For the latter, we test three options for tidal correction (see Sect. 2.2). For consistency, the timeseries from the North Sea tide gauge used for comparison are interpolated onto the non-equidistant times of acquisition

of the altimetry observations before monthly averaging. For each cell, the altimetry and tide gauge timeseries are compared by computing the linear correlation coefficient, RMSE (after subtracting the mean sea level per timeseries) and absolute linear trends. From these statistics, we choose the grid cells where altimetry observations agree best with sea level recorded by a tide gauge directly at the coast and build one timeseries by monthly averaging all observations from these cells.

## 3.2 Cross-shore changes from the intersection of land elevation data (JARKUS) with sea level

We derive timeseries of cross-shore shoreline changes as the intersection of the JARKUS topographic and bathymetric height profiles along transects with a horizontal plane at sea surface height using functions from the JARKUS Analysis Toolbox (JAT) (van IJzendoorn, 2022). Several combinations of the two intersecting surfaces result in the six solutions for cross-shore timeseries summarised in Table 2. Comparing certain solutions allows us to study 1) the geometrical influence of sea level compared to morphodynamics on the shoreline evolution, 2) the potential of altimetric sea level changes to compute shoreline positions, 3) the geometrical effect of sea level changes due to atmospheric pressure on shoreline changes, and 4) the potential geometrical impact of future sea level rise (Table 3). We assess these questions by analysing trend differences, absolute differences and RMSE between the solutions.

| Nr | Short name | JARKUS | Sea level |
|---|---|---|---|
| 1a) | TG corrected | Time variable profile | Time variable sea level from PSMSL TG including corrections |
| 1b) | TG uncorrected | Time variable profile | Time variable sea level from PSMSL TG without corrections |
| 2) | Altimetry | Time variable profile | Time variable sea level from altimetry |
| 3) | Constant sea level | Time variable profile | Constant sea level = 0 m NAP |
| 4a) | Constant profile (past) | Constant profile | Time variable sea level from PSMSL TG without corrections |
| 4b) | Constant profile (future) | Constant profile | Time variable sea level from projection until the year 2100 |

**Table 2.** The combination of constant or time variable JARKUS profiles with constant or time variable sea level from the PSMSL tide gauge (TG) or from altimetry results in six solutions for timeseries of cross-shore change.

To separate the geometrical effects of sea level and morphodynamics on the shoreline we either fix the JARKUS profile or sea level in time in order to compare these results against a version where both JARKUS and sea level are time variable. Fixing sea level to a certain height results in shoreline changes only due to morphological processes. On the other hand, fixing the JARKUS profile in time shows us the separated effect of sea level changes on shoreline evolution.

In order to learn to what extent the altimetric sea level anomalies extracted in Sect. 3.1 can be used to study shoreline evolution, we compute cross-shore timeseries with time variable sea level from the PSMSL tide gauge (TG) and from altimetry.

| Solution difference | Insights |
| --- | --- |
| TG uncorrected (1b) - Constant profile (past) (4a) | Residual is the shoreline change due to morphodynamics |
| TG uncorrected (1b) - Constant sea level (3) | Residual is the shoreline change due to sea level change |
| TG corrected (1a) - Altimetry (2) | Usability of altimetry for shoreline analysis |
| TG corrected (1a) - TG uncorrected (1b) | Influence of sea level changes due to atmospheric pressure changes |
| Constant profile (past) (4a) - Constant profile (future) (4b) | Impact of future sea level rise |

**Table 3.** Insights gained by differencing the solutions in Table 2.

The tide gauge data has been corrected for vertical land motion and atmospheric pressure. To be consistent with JARKUS, the tide gauge data is yearly averaged, making tidal correction unnecessary. The observed vertical land motion is with approximately -0.5 $\mathrm{mm\,yr^{-1}}$ (see Sect. 2.3) small. Atmospheric pressure however was on average lower than the global mean
pressure, leading to a bias in the yearly averaged IB correction between -1 cm and -4.5 cm (see Sect. 2.2) with the corrected sea level below uncorrected sea level. Therefore, we test the effects of the IB correction on the shoreline variability by computing cross-shore timeseries from time variable tide gauge data with and without corrections.

Finally, the potential geometrical impact of future sea level rise in the theoretical absence of morphodynamics is assessed by
intersecting a fixed JARKUS profile with projected sea level. We use a sea level projection for Den Helder from CMIP5 models as computed by Vermeersen et al. (2018), resulting in a cumulative sea level rise of $0.52 \pm 0.27$ m for the years 2018–2100 under RCP4.5. Here, we simplify the projected sea level rise to be linear assuming a constant rise of 0.62 $\mathrm{cm\,yr^{-1}}$ as we only look into long-term shoreline changes.

From the 217 JARKUS transects covering Terschelling, we removed 65 transects from the computation by thinning out the areas around West- and East-Terschelling where due to the curvature of the shoreline several of the provided transects cover almost the same beach section. From the remaining 152 selected transects we only use the transects that provide at least 5 years of data between 1992 and 2022. Additionally, we also excluded one transect that exhibited unrealistic jumps larger than 2000 m from all computations.

**3.3 Cross-shore changes from satellite-derived shorelines**

We compute timeseries of cross-shore changes by intersecting the shorelines extracted from optical satellite images using CASSIE and CoastSat (see Sect. 2.5) with the transect coordinates from JARKUS. The intersection computation is done with functions from the CoastSat toolbox (Vos et al., 2019b). For the CASSIE-derived shorelines, we used the same 152 transects as for the JARKUS shorelines in Sect. 3.2. Using the JARKUS transects allows us to directly compare the CASSIE-derived
shorelines against the JARKUS shorelines (see Sect. 3.5). The CoastSat estimates are limited to a region of 100 transects

around the center of the coastline due to performance issues in larger areas.

As the satellite images are taken at different tidal stages we applied a tidal correction derived from the following relationship:

$$\Delta x = \frac{\Delta h}{\tan \beta},$$
(2)


where $\Delta x$ is the horizontal shift of the shoreline due to the difference $\Delta h$ between the actual sea level at time of image acquisition and a reference sea level (here set to 0 m NAP) for a coastal section with beach slope $\tan \beta$. We use water levels from the North Sea tide gauge and from the EOT20 tidal model, and interpolate them to the points in time of image acquisition. For the beach slope, we first compute the topography gradients from the JARKUS dataset and then take the median over all gradients along each transect in a certain buffer zone (in the following called "cross-shore buffer zone") around the shoreline position. This gives us an estimate of beach slope that varies in both spatial dimensions and in time. The horizontal shift $\Delta x$ resulting from equation (2) can become unrealistically large, especially for small beach slopes. Some of the calculated beach slopes get as small as $8 \cdot 10^{15}$ $(\tan \beta)$ or even 0, leading to corrections up to 3.8 km or even infinity. We therefore apply an arbitrary threshold of $\pm$ 100 m for the maximum tidal correction. The tidally corrected cross-shore timeseries are smoothed using a moving average filter with a window size of five observations that are non-equidistant in time.



### 3.4 Sensitivity analysis of cross-shore changes from CASSIE

A sensitivity analysis is carried out to quantify the influence of four parameters involved in the computation of cross-shore timeseries from CASSIE for the full available period 1984–2022. The goal is to get an idea of the uncertainty of the satellite-derived shorelines and to make an informed choice of the parameter settings for further use.


**Choices during the computation of intersections between shorelines and transects**

First, the CoastSat function to compute the intersection between shorelines and transects offers the possibility to include a quality control procedure. This quality control applies for example thresholds for the standard deviation, the range and maximum values of the points involved in the computation, outlier rejection or the handling of several intersections along one transect. We test the influence of using the function with or without quality control to compute the intersections. Second, this function actually computes several intersections per transect in a zone with a certain along-shore length, in the following called "along-shore zone". The median of these intersections is the final cross-shore position. Our second experiment is to test 13 values between 50 m and 2500 m for the length of the along-shore zone.


**Choices for tidal correction**

Third, the influence of using a tidal correction is investigated. We compare the use of no tidal correction, a tidal correction computed with a uniform beach slope (same beach slope for each transect and for each point in time) and the tidal correction using the variable beach slopes described in the previous section. For the latter, we use six cross-shore buffer zones along a


transect between $\pm\ 5$ m and $\pm\ 105$ m. Finally, we analyse the influence of the source of water levels where we compare the use of observations from the North Sea tide gauge against the use of sea level from the tidal model EOT20.

The results from each of these four experiments are one timeseries per transect and therefore two-dimensional. As a representative one-dimensional measure we compute the differences to the median of all timeseries per transect resulting from changing one of the parameters. Additionally, we look at the influence of changing these parameters on standard deviations and trends.

## 3.5 Validation of cross-shore changes from CASSIE

We validate the tidally corrected timeseries of cross-shore changes from CASSIE derived with the settings found in Sect. 3.4 by comparing them to two other datasets, satellite-derived shorelines from CoastSat using the same settings, and shorelines derived from the intersection of JARKUS with sea level. As means of comparison we compute the absolute differences, standard deviations, trends and correlations.

For the comparison with CoastSat, we have an overlapping period with CASSIE of five years between 2015 and 2020. The CASSIE-derived shorelines were extracted only from Landsat images, whereas in CoastSat we included additionally Sentinel-2 images. As a consequence, there are a total of 183 images used in CoastSat, but only 23 images in CASSIE when reduced to the same 5-year period.

For the comparison with the JARKUS-derived shorelines, we used the cross-shore changes resulting from the intersection with the uncorrected tide gauge (solution 1b), as this solution represents best the actual shoreline evolution. Temporal matching is done by interpolating the cross-shore positions from CASSIE with their irregular times of image acquisition on the yearly JARKUS time vector, covering the a period of 30 years from 1992 to 2022.

## 4 Results

### 4.1 Nearshore sea level variability from altimetry compared to tide gauges

In terms of correlations and RMSE aggregated over all cells, the timeseries of altimetric sea level anomalies show the best similarity with the PSMSL tide gauge with an average linear correlation coefficient of 0.53 and an average RMSE of 12.3 cm (Fig. 3). The comparison to the North Sea tide gauge corrected with EOT20 yields the second highest correlation coefficient with an average of 0.42 and the third lowest RMSE with an average of 19.3 cm. Almost the same results are achieved when correcting the tide gauge with FES2014 (not shown). Filtering the North Sea tide gauge data with a 30-day Butterworth filter leads to reasonable results with correlations around 0.4 and up to 0.6 and an average RMSE of 14.5, while correcting with the

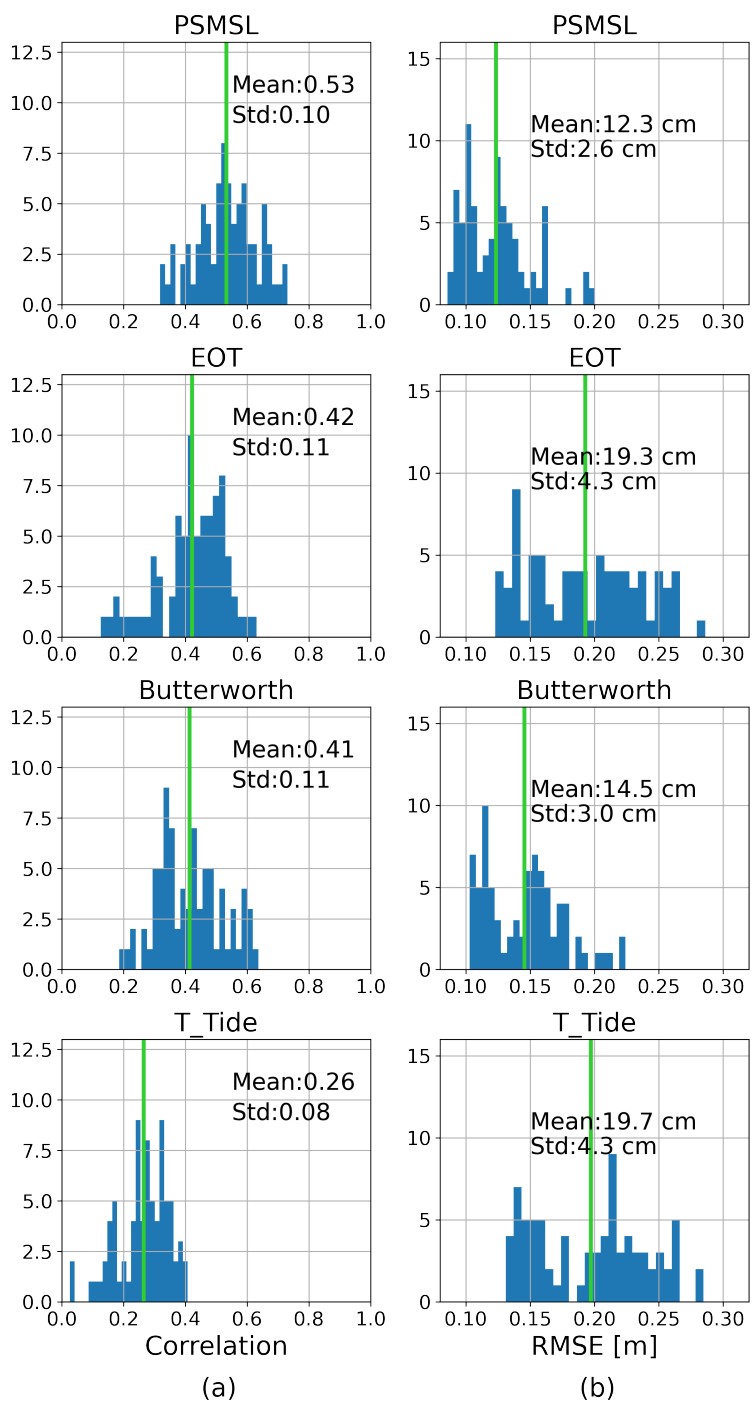

**Figure 3.** Histograms of linear correlation coefficients (a) and RMSE (b) between sea level anomalies altimetry and from tide gauges per cell for different tide gauge solutions, from top to bottom: Monthly PSMSL tide gauge without tidal correction, 10-minute resolution North Sea tide gauge corrected using the EOT tidal model, the North Sea tide gauge filtered with a 30-day Butterworth filter and the North Sea tide gauge corrected with T_Tide. The green vertical lines indicate the mean.

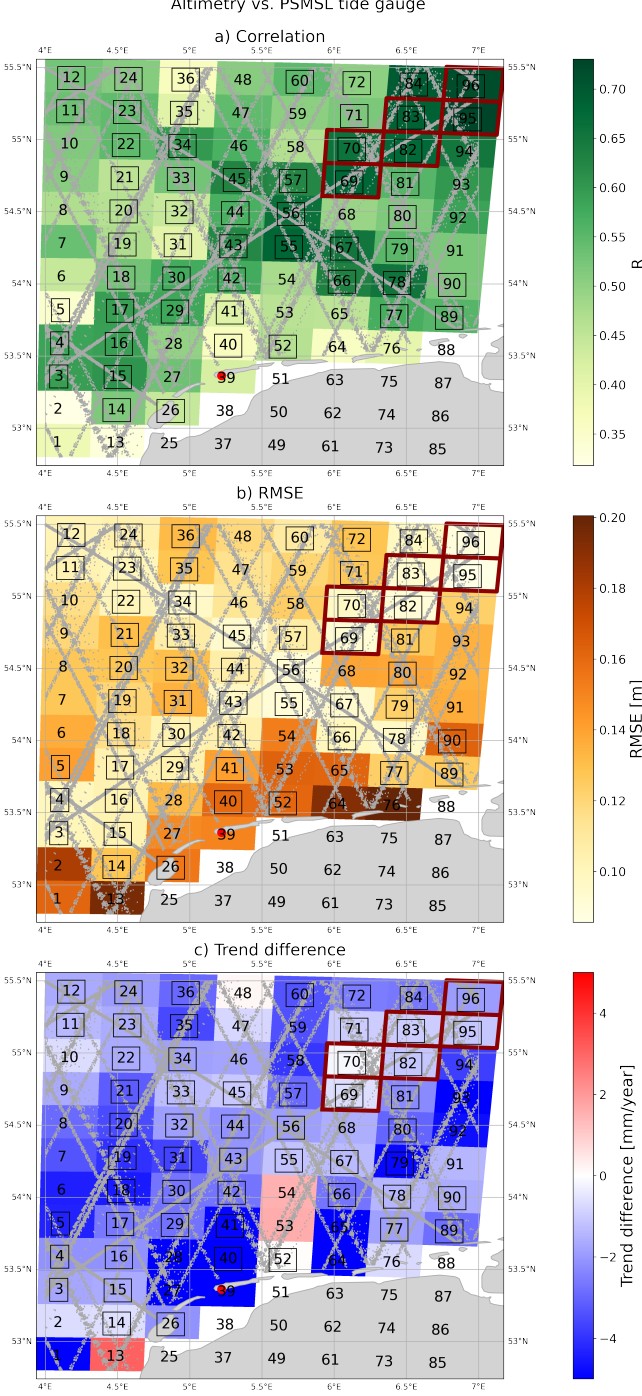

**Figure 4.** Maps of linear correlation coefficients (a) and RMSE (b) between sea level anomalies from altimetry and from the PSMSL tide gauge per cell. (c) Map of differences between the linear trends from the altimetry timeseries and from the PSMSL tide gauge. The numbers indicate the cell numbers. Cell numbers surrounded by a black box contain timeseries that cover at least the period 01.07.2002–31.03.2020. Cells surrounded by a red box are selected to create a timeseries of monthly altimetric sea level anomalies for further use.

estimated tidal signal from T_Tide yields no significant correlation at all.


Creating altimetry timeseries from cells allows us to study the spatial distribution of correlation, RMSE and trend differences. As an example, the maps in Fig. 4 show the comparison of each altimetric sea level anomaly timeseries with the PSMSL tide gauge. There is a clearly visible pattern of higher correlation coefficients with values between 0.6 and 0.7 and lower RMSE with values between 9 and 11 cm in a wide cross over the tracks from the JASON satellites. Along these tracks, the altimetry

timeseries cover the entire 18-year period, whereas timeseries in other locations are often significantly shorter.

Differencing the altimetric sea level trends with the trend from the PSMSL tide gauge results in a more scattered pattern. The absolute linear sea level trend for the period January 2002–April 2020 as observed by the PSMSL and the North Sea tide gauge is 4.7 $\mathrm{mm\,yr^{-1}}$ and 4.9 $\mathrm{mm\,yr^{-1}}$, respectively. The differences between altimetry and PSMSL tide gauge are regularly

in the order of magnitude of observed sea level trends. This is another indication that the nearshore altimetry measurements cannot always properly represent sea level variations at the coast. However, the differences along the JASON tracks are again smaller with values below 1.5 $\mathrm{mm\,yr^{-1}}$.

The tide gauge data are corrected for atmospheric pressure using ERA5 data (see Sect. 2.2), whereas the altimetry data from

OpenADB were corrected with the Dynamic Atmospheric Correction (DAC) by Carrère and Lyard (2003). However, when we apply the DAC to the tide gauge data, we find no similarity in terms of correlation and RMSE with the altimetry timeseries, mainly caused by two peaks in the DAC dataset that are not present in the altimetry dataset. Moreover, de-trending and de-seasoning the timeseries removed all correlations, therefore all similarities are caused only by the inter-annual signal.

For extracting an altimetry sea level timeseries for further use that is representative for sea level change at the shoreline we consider only the cells that cover at least the period 01.07.2002–31.03.2020 (213 months). From a total of 80 cells, this leaves 54 cells that contain a total of 125–220 months of data. The 10 best scoring cells in terms of linear correlation coefficient, RMSE and the trend difference to the PSMSL tide gauge in Table A1 indicate two regions as possible candidates. One lies in the North East of the studied region about 250 km away from the coast (cells 96, 95, 83, 82, 70, 69), the other one lies more in

the center about 125 km away from the coast (cells 55, 67, 78). As correlation and RMSE are slightly better for the North East region (see Tables A2 and A3 for the statistics of both regions), we continue to work with a timeseries generated from these six cells. The resulting timeseries has an absolute linear sea level trend of 3.6 $\mathrm{mm\,yr^{-1}}$ for the period January 2002–April 2020.

## 4.2 Cross-shore changes from the intersection of JARKUS with sea level

The Terschelling shoreline exhibits retreating and advancing areas (Fig. 5). From these we select three sections, two on the

outer retreating parts of the shoreline (sections A and C), and one on the central advancing part of the shoreline (section B). Larger trends appear in areas with mild beach slopes, while small trends are related to steeper parts of the beach (see Fig. A3).

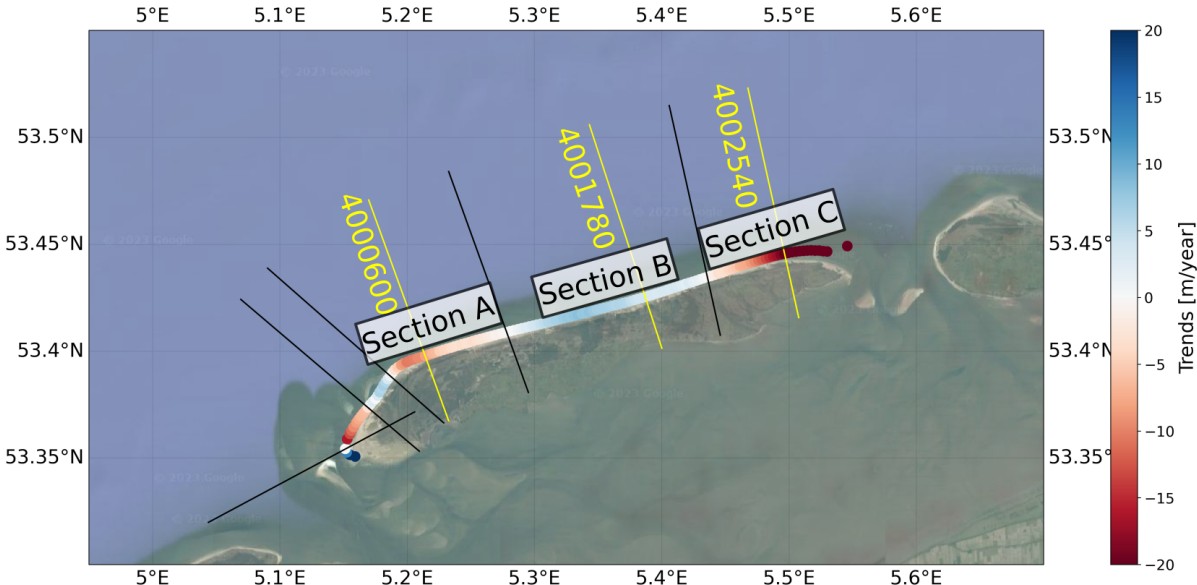

**Figure 5.** Trends of shoreline change derived from the JARKUS datasets differ along the coast of Terschelling, showing regions of seaward and landward movements. The trends shown here are computed by intersecting the profiles with the uncorrected PSMSL tide gauge (solution 1b). The black lines indicate the transects at the transitions between retreating and advancing sections of the coast. For the further analysis we focus on the two retreating sections A and C on the outer parts of the barrier island and the advancing section B approximately in the middle of the coastline. Furthermore, the transects used to extract the timeseries in Fig. 6 are shown in yellow. Background image from Google Map tiles using cartopy.io.img_tiles (© Google Maps)

Three examples of cross-shore timeseries resulting from the intersection computation for the solutions (1) - (4a) from each section are given in Fig. 6. The respective transects are indicated in Fig. 5. For these example transects, all of the solutions (1) - (3) with time variable JARKUS profiles show very similar shoreline changes. In contrast, solution (4) with the JARKUS profile fixed in 1992 does hardly show any visible shoreline variability.

In the following, we compare the shoreline trends from the different solutions from Tables 2 and 3 in terms of linear trends, absolute differences and RMSE. An overview of the statistics, averaged over the transects along the entire coastline and for the respective sections A, B and C, is given in the Tables 4 (trends), 5 (absolute differences) and 6 (RMSE). Overall, we find that the RMSE is always significantly larger than the respective absolute differences. It is therefore difficult to draw conclusions about an offset in absolute position between two compared shorelines. However, we can analyse the trend differences to answer the following questions:

**What is the geometrical influence of sea level compared to morphodynamics on the shoreline evolution?**

Under the assumption that the beach profile does not change over time (solution 4a), all sections respond with retreating shore-

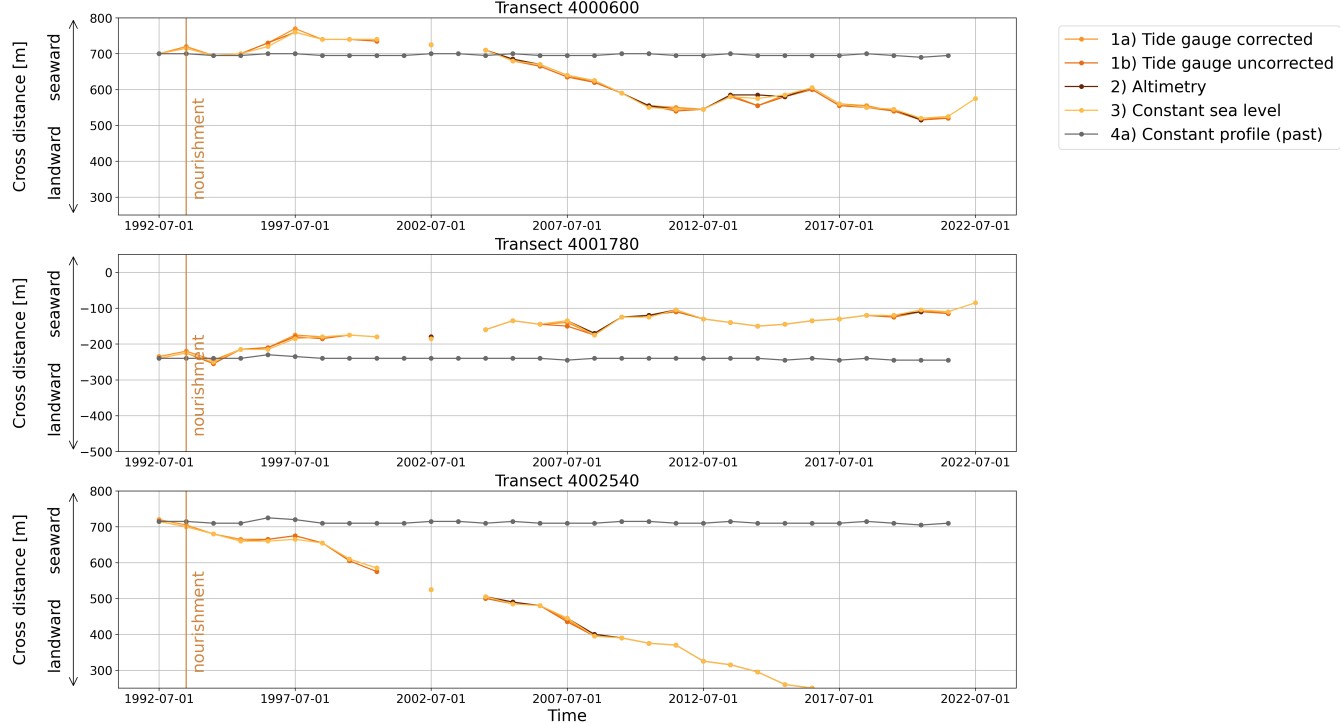

**Figure 6.** Three examples of cross-shore timeseries for the transects indicated in Fig. 5 showing landward trends in West and East Terschelling, and seaward trends for the central coast. The cross-distance on the y-axis is defined to point landward in the negative direction, while more positive values indicate a movement seawards. A negative trend therefore can be interpreted as a retreating shoreline, whereas a positive trend indicates an advancing shoreline. The absolute values depend on the definition of the coordinate system along the transects and can not be compared.

line trends. With magnitudes between -0.2 $\mathrm{m\,yr^{-1}}$ and -0.3 $\mathrm{m\,yr^{-1}}$, inundation by sea level change is however rather small when compared to solution 1b) with time variable profiles where the landward trend in sections A and C ranges between -4.7 $\mathrm{m\,yr^{-1}}$ and -18.1 $\mathrm{m\,yr^{-1}}$, respectively.


When comparing trends of shoreline changes resulting from variable sea level (1b) to the shorelines resulting from constant sea level 0 m over NAP (3) we find negative trend differences of -0.4 $\mathrm{m\,yr^{-1}}$ and -0.2 $\mathrm{m\,yr^{-1}}$ for sections A and B, but positive differences of 0.3 $\mathrm{m\,yr^{-1}}$ for section C and when averaged over the entire coastline. This indicates that timeseries from variable sea level (1b) in the central part of the coastline exhibit on average more landward trends, whereas at the outer parts

of Terschelling the timeseries from variable sea level (1b) have larger seaward trends.

**Can we use altimetric sea level changes to compute shoreline positions?**

The trend differences between solution (1a) using the corrected PSMSL tide gauge and solution (2) using sea level from al-

timetry are always equal or smaller than $0.1\,\mathrm{m\,yr^{-1}}$. Consequently, exchanging the tide gauge data with an estimate based on altimetry is justified.

**To what extent do sea level changes due to atmospheric pressure alter shoreline changes?**

When comparing the solution using the corrected PSMSL tide gauge (1a) to the uncorrected PSMSL tide gauge (1b), sections B and C exhibit small trend difference of (-)$0.1\,\mathrm{m\,yr^{-1}}$. In terms of trends there is therefore no detectable difference induced by the correction for atmospheric pressure. We would expect a bias in shoreline position as the IB correction lowers the sea level by 1 to $4.4\,\mathrm{cm}$, moving the shoreline seawards. This effect can be observed in the absolute differences ranging between 2.2 and $2.9\,\mathrm{m}$, but due to the RMSE between 4.2 and $6.4\,\mathrm{m}$ a bias is not detectable with sufficient certainty.

**What is the potential geometrical impact of future sea level rise?**

We did the computation with the profile fixed in 1992 twice, first intersecting it with sea level from the uncorrected PSMSL tide gauge over the period 1992–2022, and second intersecting it with projected sea level over the period 1992–2100. For the three sections as well as for the entire shoreline, the landward trend increases for the 108 year period by $0.2$–$0.3\,\mathrm{m\,yr^{-1}}$ compared to the 30 year period.

| Trends [$\mathrm{m\,yr^{-1}}$] | Entire coastline | Section A | Section B | Section C |
|---|---|---|---|---|
| 1a) PSMSL TG corrected | -3.4 ± 1.2 (0.9) | -4.7 ± 2.1 (0.9) | 4.3 ± 1.0 (0.9) | -18.0 ± 1.3 (1.0) |
| 1b) PSMSL TG uncorrected | -3.4 ± 1.2 (0.9) | -4.7 ± 2.1 (0.8) | 4.4 ± 1.0 (0.9) | -18.1 ± 1.2 (1.0) |
| 1a) PSMSL TG corrected, reduced to altimetry period | -3.3 ± 0.7 (0.8) | -4.6 ± 1.4 (0.6) | 3.2 ± 0.6 (0.9) | -16.7 ± 0.9 (1.0) |
| 2) Altimetry | -3.4 ± 0.7 (0.8) | -4.6 ± 1.4 (0.6) | 3.2 ± 0.6 (0.8) | -16.8 ± 0.8 (1.0) |
| 3) Constant sea level | -3.7 ± 1.2 (0.9) | -4.4 ± 2.1 (0.9) | 4.5 ± 1.0 (0.9) | -18.3 ± 1.3 (1.0) |
| 4a) Constant profile (past) | -0.3 ± 0.2 (0.8) | -0.2 ± 0.2 (0.7) | -0.3 ± 0.3 (0.8) | -0.3 ± 0.3 (0.9) |
| 4b) Constant profile (future) | -0.5 ± 0.1 (1.0) | -0.5 ± 0.1 (1.0) | -0.5 ± 0.1 (1.0) | -0.6 ± 0.1 (1.0) |
| **Trend differences [$\mathrm{m\,yr^{-1}}$]** | | | | |
| TG uncorrected (1b) - Constant profile (past) (4a) | -3.1 | -4.6 | 4.6 | -17.7 |
| TG uncorrected (1b) - Constant sea level (3) | 0.3 | -0.4 | -0.2 | 0.3 |
| TG corrected (1a) - Altimetry (2) | 0.1 | 0.0 | 0.0 | 0.1 |
| TG corrected (1a) - TG uncorrected (1b) | 0.0 | 0.0 | -0.1 | 0.1 |
| Constant profile uncorrected PSMSL (4a) - Sea level projection (4b) | 0.2 | 0.3 | 0.2 | 0.3 |

**Table 4.** Trends and trend differences in $\mathrm{m\,yr^{-1}}$ for all investigated solutions described in tables 2 and 3, averaged over all transects along the entire coastline, and over the sections A (West), B (Center) and C (East). The error margins represent the 5-95 % confidence interval. Numbers in brackets indicate the averaged significance of the trends according to the Mann-Kendall test (1: Trend is significant within the 5-95 % confidence interval, 0: No significant trend). Time periods: 1992–2022 for solutions 1a)-1d), 3) and 4a, 2004–2021 for solution 2), 1992–2100 for solution 4b).

| Absolute differences [m] | Entire coastline | Section A | Section B | Section C |
| --- | --- | --- | --- | --- |
| TG uncorrected (1b) - Constant profile (past) (4a) | -38.4 | -64.9 | 89.7 | -234.5 |
| TG uncorrected (1b) - Constant sea level (3) | -1.4 | -1.2 | -1.8 | -1.4 |
| TG corrected (1a) - Altimetry (2) | -0.5 | -0.5 | -0.6 | -0.5 |
| TG corrected (1a) - TG uncorrected (1b) | 2.5 | 2.2 | 2.6 | 2.9 |

**Table 5.** Absolute differences in m between some of the solutions, averaged per transect over time and averaged over all transects along the entire coastline, as well as over the sections A (West), B (Center) and C (East).

| RMSE [m] | Entire coastline | Section A | Section B | Section C |
| --- | --- | --- | --- | --- |
| TG uncorrected (1b) - Constant profile (past) (4a) | 149.5 | 91.1 | 104.4 | 284.4 |
| TG uncorrected (1b) - Constant sea level (3) | 5.6 | 4.4 | 5.9 | 5.5 |
| TG corrected (1a) - Altimetry (2) | 2.9 | 3.1 | 3.0 | 2.8 |
| TG corrected (1a) - TG uncorrected (1b) | 5.3 | 4.2 | 5.6 | 6.4 |

**Table 6.** RMSE in m for combinations of some of the solutions, averaged over all transects along the entire coastline, as well as over the sections A (West), B (Center) and C (East).

## 4.3 Sensitivity analysis of cross-shore changes from CASSIE

**Choices during the computation of intersections between shorelines and transects**

We first test the effect of computing intersections between shorelines and transects on the resulting cross-shore timeseries with or without the quality control implemented in CoastSat. In trends (Fig. 7) and in standard deviations (not shown), we find that using the function with quality control produces more stable results in the outer parts of the shoreline. These areas are characterised by ebb tidal deltas between the Wadden Sea and the North Sea with several shoals, spits and tidal flats (see Fig. 1) where probably several intersections are found. We therefore suspect that the function with quality control selects the intersections that belong to the main shoreline. When looking at the absolute differences between each timeseries (no quality control - with quality control) we find an average bias of -115.0 m, therefore the shorelines computed without quality control are on average further inland. However, this difference increases from -15.0 m to -226.5 m for along-shore zones from 50 m to 2500 m.

Second, when testing different values for the length of the along-shore zone, the timeseries with quality control have on average a higher variability with standard deviations up to 120 m, whereas the timeseries without quality control range around 80 m standard deviation (see Fig. A4). Average trends are with values between -1.7 $\mathrm{m\,yr^{-1}}$ and -2.0 $\mathrm{m\,yr^{-1}}$ more stable for the non-quality controlled timeseries, while the trends of quality controlled timeseries decrease considerably for along-shore

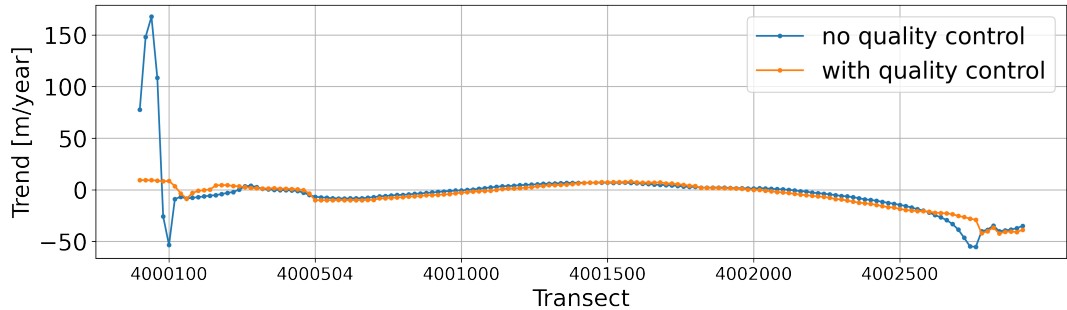

**Figure 7.** CASSIE shoreline trends for each transect from the CoastSat functions to compute the intersection between shoreline and transect without and with quality control. The length of the along-shore zone is here fixed to 1200 m. The six transects on the tip of the Western part of the island show strong deviations in the average trend in the non-quality controlled version that are not evident in the quality controlled version.

zones higher than 1500 m up to -3.6 $\mathrm{m\,yr}^{-1}$. The non-quality controlled timeseries are also more stable with regard to the difference to the median of all solutions that reach a maximum of -1 m. In contrast, the differences of the quality-controlled timeseries show a linear decrease from 84.3 m to -67.5 m with the zero-crossing at 1250 m along-shore zone length.

We continue to compute the intersections with the quality control function as the cross-shore trends are more consistent along the shoreline. For the length of the along-shore zone we continue to work with 1200 m, making a trade-off between higher standard deviation but smaller difference to median, while keeping the trends at a reasonable magnitude.

**Choices for tidal correction**

The effect of using a tidal correction on the shoreline position is illustrated in Fig. 8 for the three example profiles that were earlier shown for the cross-shore changes from JARKUS in Sect. 4.2. Our third experiment is to test different tidal corrections, using a uniform beach slope with $tan\beta = -0.01$ or using a variable beach slope (variable in along- and cross-shore direction, as well as in time) with cross-shore buffer zones between $\pm$ 5 m to $\pm$ 105 m around the shoreline position. The variability of the corrected timeseries is reduced by all tested types of tidal corrections to standard deviations (given as the median over all transects) between 87.5 m and 88.2 m, compared to 152.5 m for the uncorrected timeseries (see Fig. A5). Trends for the timeseries corrected with variable beach slopes vary between -0.4 $\mathrm{m\,yr}^{-1}$ and -1.2 $\mathrm{m\,yr}^{-1}$, where larger cross-shore buffer zones tend to lead to larger trends. Using the uniform beach slope leads to seaward trends of -1.4 $\mathrm{m\,yr}^{-1}$, using no tidal correction results in a trend of -1.6 $\mathrm{m\,yr}^{-1}$. The difference to the median of all solutions is small with a maximum of 0.2 m for all tested tidal corrections with variable beach slopes, whereas the uniform beach slope leads to a deviation of 1.7 m. Not applying a tidal correction leads to a difference of -2.4 m. As applying a tidal correction reduces the standard deviation and the results are closer to the solution-median, we continue to work with the tidal correction using a variable beach slope with a 45

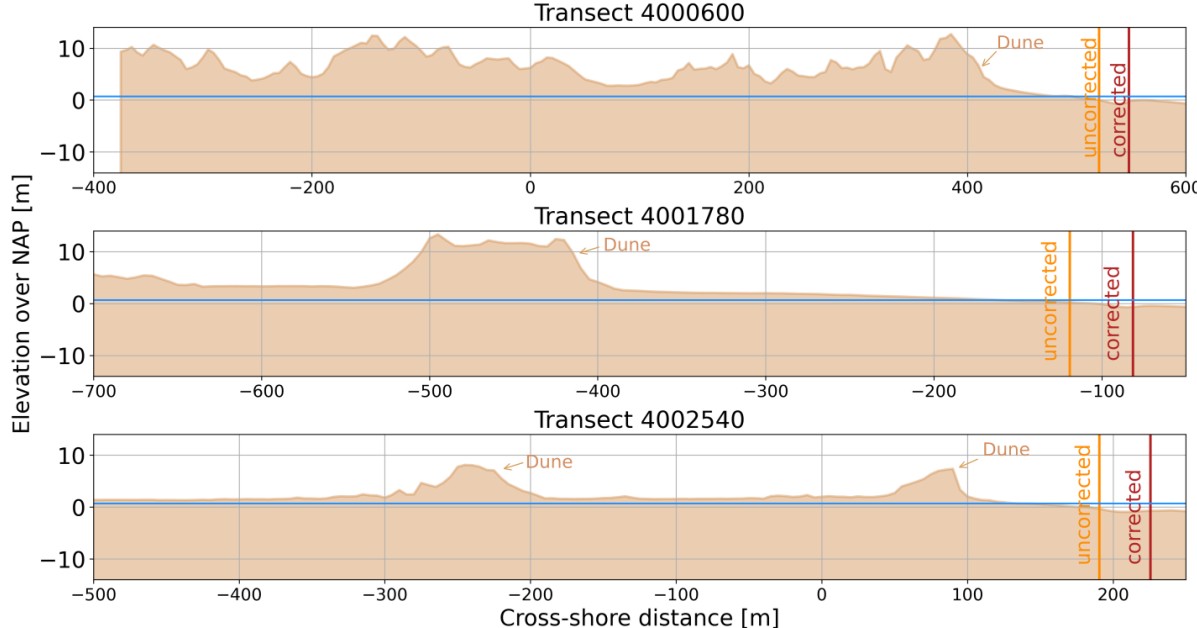

**Figure 8.** JARKUS profiles for the three example transects in Fig. 5 for the year 2020. The vertical lines indicate the CASSIE shoreline positions at 28.05.2020 10:32:54, without tidal correction (orange) and with tidal correction (brown). The tidal correction was computed using time- and space-variable beach slopes with a ± 45 m cross-shore buffer zone. The blue horizontal line indicates the water level (+67.7 cm) from the uncorrected North Sea tide gauge at the time of image acquisition. The reference sea level used for tidal correction is 0.00 m (see also Sect. 3.3).

m cross-shore buffer zone.

Lastly, we compare the effect of using different sources of water levels for tidal correction, using measurements from the North Sea tide gauge and estimates from the EOT20 tidal model. When subtracting timeseries corrected with EOT20 from timeseries corrected with the tide gauge, we find differences that are almost constantly negative between -2.0 m and -14.3 m across all transects (Fig. A6). Consequently, the cross-shore changes tidally corrected with tide gauge observations are on average more landward than the timeseries using water levels from EOT20. These differences have a tendency to get larger towards the Eastern part of the shoreline. However, the source of water levels does not have an impact on the trends. We will continue to compute tidal corrections using the tide gauge observations.

## 4.4 Comparison of cross-shore changes from CASSIE and from CoastSat

When comparing standard deviations and trends per transect of cross-shore changes between CASSIE and CoastSat over the available 5-year period, the mismatch in sampling points with 23 images in CASSIE and 183 images in CoastSat manifests in large deviations for CASSIE over a wide part of the coast with standard deviations up to 200 m and trends down to -80 m yr$^{-1}$.

These discrepancies in CASSIE do not appear over the entire 40-year period used in Sect. 4.3 where a single discontinuity in the cross-shore timeseries has less impact. However, when we aggregate all absolute differences between both timeseries for all 100 transects covered by CoastSat in histogram (Fig. 9), we see a negative bias of -39.2 m on average. This result indicates that the CASSIE-derived shorelines have a tendency to lie further seaward than the shorelines extracted with CoastSat.

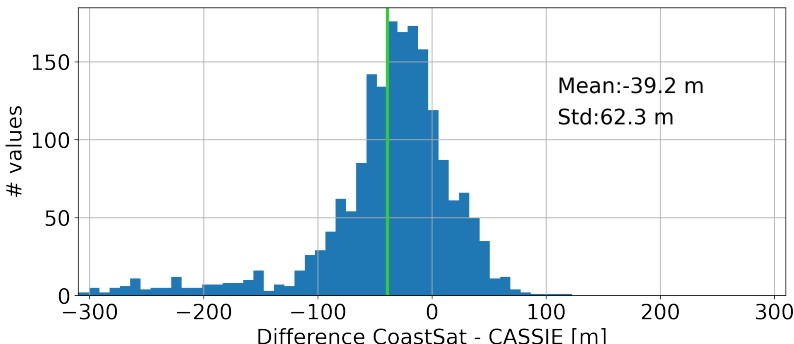

**Figure 9.** Histogram of absolute differences between the cross-shore timeseries from CoastSat minus CASSIE, both including tidal correction. The vertical green line indicates the mean. On average, shoreline positions derived with CASSIE are more landward compared to shorelines from CoastSat.

## 4.5 Comparison of cross-shore changes from CASSIE and from JARKUS

We assess the agreement between cross-shore changes extracted from Landsat images with CASSIE ("CASSIE-derived shorelines") and cross-shore changes computed as the intersection of JARKUS profiles with sea level ("JARKUS shorelines") in terms of their absolute differences, their standard deviations and trends, as well as the correlation per transect (Fig. 10). The absolute differences show a clear bias of -82.8 m on average where the CASSIE-derived shorelines are for the majority of the transects further seaward. The differences are larger for the outer transects in the western and eastern curvatures of the coastline.

In terms of standard deviations and trends, both methods to derive shorelines produce similar results. On average, the standard deviation of the JARKUS shorelines is 8.9 m smaller. However, the spatial pattern with a tendency to smaller standard deviations in the middle part of the coastline and larger variations in the outer parts is very similar. The trend differences show that JARKUS shorelines are on average by 2.3 $\mathrm{m\,yr^{-1}}$ more retreating, where the larger differences appear again in the two outer parts where beach slopes are mild. Areas with large trend differences are not related to areas with bigger or smaller seaward or landward trends (see Fig. 5). Along the central part of the shoreline where the beach slope is steeper, the differences in trends are usually below $\pm\,2\,\mathrm{m\,yr^{-1}}$.

Linear correlation coefficients between timeseries of CASSIE-derived shorelines and timeseries of JARKUS shorelines show an overall reasonable similarity with an average value of 0.55. In zones with large seaward or landward trends (see Fig. 5),

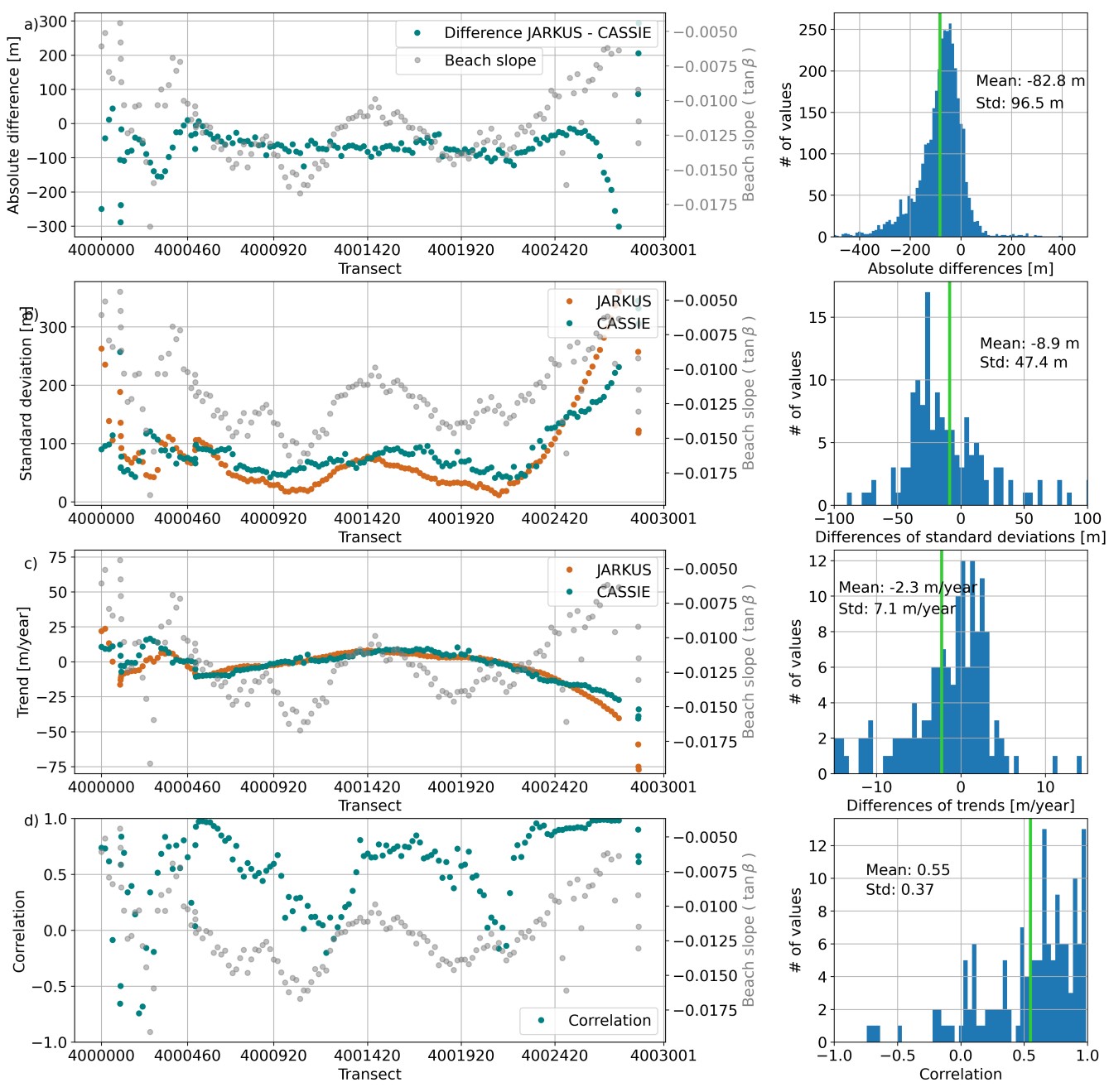

**Figure 10.** Statistics to compare timeseries of cross-shore changes from JARKUS and from satellite-derived shorelines extracted with CASSIE. a) Absolute differences JARKUS-CASSIE between the timeseries, as the median over each transect and as a histogram of all differences. b) Standard deviation per transect and histogram of differences of standard deviations (JARKUS-CASSIE). c) Trends per transect and histogram of trend differences (JARKUS-CASSIE). d) Linear correlation coefficient between CASSIE and JARKUS per transect and histogram of correlations. The vertical green lines in the histograms indicate the mean.

correlation reaches values above 0.70 and up to 0.99. Lowest correlations appear in the two stable zones, as well as in the Western curvature of the coastline.

## 5 Discussion and conclusions

A combined analysis of observational datasets describing sea level variations in relation to shoreline changes allows us to draw conclusions about the geometrical effect of long-term relative sea level changes and morphodynamics on shoreline movements
at Terschelling. Furthermore, by comparing estimates that capture similar processes we can illustrate the uncertainties in the respective datasets.

### Geometrical influence of sea level rise on shoreline changes

By intersecting topographic and bathymetric profiles from JARKUS data fixed in time with time-variable sea level we found that the observed 10.5 cm of relative sea level rise between 1992 and 2022 had a rather moderate impact on the shoreline evolution with an average landward trend of -0.3 $\mathrm{m\,yr^{-1}}$. Instead, the observed shoreline movements were mainly associated with changes in the JARKUS profiles, leading to trends in shoreline position around -3.4 $\mathrm{m\,yr^{-1}}$. From this we conclude that shoreline changes at the North Sea coast of Terschelling are currently largely driven by morphological processes, in this case
erosion. For the Wadden Sea basin, Wang et al. (2012) showed that sedimentation rates compensate relative sea level rise. However, this equilibrium is likely to be disrupted when an unknown critical rate of sea level rise is reached (Wang et al., 2012). For a predicted total sea level rise of 0.52 m for the years 2018–2100 and under the assumption that there are no morphological changes, we find an average landward trend of -0.5 $\mathrm{m\,yr^{-1}}$ and a total shoreline change of -32.4 m for the North Sea coast of Terschelling. These values are relatively small compared to the changes in the observation period, but they are
detectable. Additionally, erosion is known to be enhanced by sea level rise, for example leading to higher wave energy (e.g. D'Anna et al., 2021) and intensified tides (e.g. Jordan et al., 2021). Usually, relative sea level change by vertical land motion should be considered as well; however, in the case of Terschelling the observed VLM rates were below 1 $\mathrm{mm\,yr^{-1}}$ and had no detectable influence on the shoreline position.

### 610 Usability of altimetric sea level anomalies at the coastline

In order to get the longest possible timeseries of altimetric sea level anomalies we combined observations from several altimetry missions in grid cells and assessed their similarity with the two tide gauges in terms of temporal variations dependent on the location of the grid cell. The overall similarity between altimetry and the PSMSL tide gauge is low expressed by corre-
615 lation coefficients between 0.3 and 0.7, an RMSE between 0.09 m and 0.10 m and a median trend difference of 1.8 $\mathrm{mm\,yr^{-1}}$, compared to previous studies (e.g. Mangini et al., 2022; Wöppelmann and Marcos, 2016; Cheng et al., 2012). However, the similarity between altimetry and tide gauges appears to have strong regional dependencies. Previous studies in the North Sea

using various products (Dettmering et al., 2021; Birol et al., 2017; Cipollini et al., 2017) found maximum correlations of up to 0.8, 0.7 and 0.4 respectively, with decreasing values along the coasts of the Netherlands and France. In this context, we decided to extract an altimetry timeseries for further use from a field of approximately 50x100 $\mathrm{km}$ along the Jason tracks where correlation between altimetry and the PSMSL tide gauge ranges between 0.62 and 0.70, RMSE lies between 0.09 $\mathrm{m}$ and 0.10 $\mathrm{m}$ and trend differences are between -0.2 $\mathrm{mm\,yr^{-1}}$ and -1.9 $\mathrm{mm\,yr^{-1}}$.

We observe that altimetric sea level anomalies have a tendency to become more representative of coastal sea level the further away they are from the coast. This phenomenon was reported in several earlier publications (e.g. Cazenave et al., 2022; Birol et al., 2017; Cipollini et al., 2017), and can be explained by the known problems of altimetry in the vicinity of the coast that require the use of retracking algorithms and specialised range corrections. An unexpected finding from our experiments is that the similarity is highest with the PSMSL tide gauge situated in the harbour in the Wadden Sea at West Terschelling. In contrast, the tide gauge at the North Sea coast is spatially closer to the altimetry measurements, but led to lower similarities for all tested tidal corrections. We also expected to see higher coincidence with a tide gauge timeseries corrected with the same tidal model that was used to correct the altimetry observations, FES2014. However, tide gauge timeseries corrected with FES2014 or EOT20 scored only second in terms of correlation and third in terms of RMSE, implying that global tidal models still can not capture significant tidal signals at the coast. A means of improvement might be to use the new regional tidal model EOT-NECS by Hart-Davis et al. (2023). Correcting the tide gauge by removing the observed tidal frequencies with T_Tide resulted in lowest correlations and second-lowest RMSE.

Replacing the tide gauge data in the shoreline computation from JARKUS with the extracted altimetry timeseries resulted in the same shoreline trends. We conclude that the uncertainties in the altimetric sea level anomalies do not hinder their use for shoreline analysis, which opens the possibility to study the influence of sea level on shorelines that are not covered by tide gauges.

**Reliability of satellite-derived shorelines from CASSIE**

For the extraction of shorelines from optical satellite images we mainly relied on the software CASSIE with Landsat images, yielding timeseries of cross-shore changes over 39 years between 1984 and 2023. Almeida et al. (2021) compared their outcomes of CASSIE using Landsat 8 images to in-situ GNSS observations in one week during summer along four sandy beaches in Brazil, and found with an RMSE of 8.84 $\mathrm{m}$ a similar range of uncertainties as previous studies of satellite-derived shorelines. In order to learn more about the uncertainties of the CASSIE-derived shoreline timeseries at Terschelling, we tested their sensitivity to tidal correction and parameters involved in the computation of timeseries. For validation, we compared the CASSIE-derived shorelines to satellite-derived shorelines from the CoastSat toolbox by Vos et al. (2019b), as well as the shorelines computed from the intersection of JARKUS with sea level.

When correcting the cross-shore positions for tides, we found that using a non-uniform beach slope that varies in along-shore and cross-shore direction as well as in time considerably improved the results. Applying this tidal correction reduced the temporal variability on average by up to 62 m and the difference to the median of all solutions by about 2 m compared to using no tidal correction. Additionally, we saw an average trend increase by about $0.8 \, \mathrm{m} \, \mathrm{yr}^{-1}$. We hypothesise that this difference in long-term trends is caused by the sampling interval due to the sun-synchronous orbits of the Landsat satellites, leading to aliasing of certain tidal frequencies (e.g. Eleveld et al., 2014; Bishop-Taylor et al., 2019b).

Using a uniform beach slope that is constant in space and time, as was done in previous publications (e.g. Chen and Chang, 2009; Vos et al., 2019a; Adebisi et al., 2021), reduced the standard deviation compared to using no correction but yielded with 1.7 m a higher difference to the median of all solutions and higher trends. However, we only tested a single beach slope, and the results could differ for other values. We conclude that especially for a coast with very mild beach slopes, it is preferable to compute the horizontal shoreline shift due to tides with a space- and time-variable beach slope, although we realize that this information is not always available.

Apart from the tidal correction, other parameters in the computation of cross-shore timeseries from satellite-derived shorelines showed their potential to alter the results considerably. For the upper and lower limit of the different tested settings, the absolute shoreline position changed up to 226 m, while trends differed up to $1 \, \mathrm{m} \, \mathrm{yr}^{-1}$. We expect that there are different best settings for different sites, dependent for example on the degree of curvature of the shoreline or the presence of shoals and other features that, from space, look similar to a shoreline.

Due to difficulties with cloud masking in the CoastSat toolbox, we only had results from a relatively short time period of five years between 2015 and 2020 for a limited part around the central shoreline at hand. The comparison between CASSIE and CoastSat showed large deviations in trends and standard deviations that can be mainly attributed to the low number of Landsat images used in CASSIE in that period, increasing the impact of single outliers. We conclude therefore that trends from the presented CASSIE-derived shoreline changes are only reliable over longer time periods. More insights might be gained by including Sentinel-2 images in the CASSIE computation, however it should be tested before what the effect of combining Landsat surface reflectances and Sentinel-2 Top-of-Atmosphere reflectances in one timeseries will be. While we cannot draw any conclusions about the uncertainties in shoreline trends due to the low number of images in the solution from CASSIE, we did observe a bias in absolute shoreline positions, where the CASSIE-derived shorelines are on average 39.2 m further seaward than the shorelines from CoastSat. A recent study by Vos et al. (2023) comparing five state-of-the-art shoreline detection algorithms (including CASSIE and CoastSat) finds that the accuracy of absolute biases and long-term trends depends on the hydrologic and morphologic setting of the study area, with lower accuracies at sites with higher tidal range, higher wave energy, smaller beach slopes and more complicated morphology.

When comparing timeseries of CASSIE-derived shorelines with timeseries of JARKUS shorelines, we first note a bias of 82.8 m on average, where the CASSIE-derived shorelines are further seawards than the JARKUS shorelines. The linear correlation coefficients between both estimates are high with values between 0.70 and 0.99 in regions with higher landward or seaward trends, but reach also low and negative values below 0.50 in regions without a clear shoreline trend. Additionally, we found a difference in shoreline trends where the JARKUS shorelines were on average $2.3 \, \mathrm{m \, yr^{-1}}$ more retreating with larger deviations in the Eastern and Western parts of the shoreline. These outer parts are characterised by small beach slopes, a stronger shoreline curvature and additional seaward morphological features like shoals and spits, which hamper shoreline detection from satellite images. Our results are in line with the findings in Do et al. (2019), who compared tidally corrected satellite-derived shorelines from 13 Landsat images in the period 1985–2010 with JARKUS shorelines derived from the intersection with time variable sea level from a nearby tide gauge over a coastal section of 60 km south of Texel, the most southern Wadden island (see inlay in Fig. 1). In terms of correlations between timeseries of Landsat and JARKUS shorelines, Do et al. (2019) find similar high values above 0.78 for certain zones. Although they too find a bias where the satellite-derived shorelines are further seawards, the magnitude of this bias is with 8 to 9 m on average about a factor 10 smaller than our result.

**Limitations**

This works presented an overview over different datasets used for coastal monitoring and their combined processing, at the cost of not going in depth into the details of the single techniques.

For deriving an altimetry timeseries, we restricted ourselves to the use of one single dataset. This is an along-track product retracked with ALES, an algorithm specifically designed for coastal areas, provided by the OpenADB (see section 2.1). This OpenADB ALES product has been used successfully before in studies combining altimetry and tide gauges (e.g. Mangini et al., 2022; Oelsmann et al., 2021). Our comparison to the local tide gauges and the use for computing the Jarkus shorelines in comparison to the other solutions showed that offshore altimetry can be used to study shoreline changes. However, in order to get the full picture of uncertainties in altimetry datasets, it could be useful to additionally include other products, such as the ESA Sea Level Climate Change Initiative gridded product (Copernicus Climate Change Service, 2018).

When correcting the tide gauge observations in order to make them comparable to altimetry, we applied only a correction for atmospheric pressure changes, neglecting sea level changes due to wind. Wind and atmospheric pressure are in sea level studies often accounted for by using the Dynamic Atmospheric Correction (DAC) by Carrère and Lyard (2003). However, when we integrated the DAC dataset in our calculation we found two spikes that are not exhibited in the altimetry dataset, and therefore decided not to use it. The comparison between altimetry and tide gauges could therefore be improved by finding a way to account for sea level changes due to wind in the tide gauge observations.

Another correction applied to the tide gauges for the comparison with altimetry was the vertical land motion (VLM). Here we used only data from a GNSS station as a proxy for VLM. However, this approach may neglect other ongoing processes such

as sediment compaction below the base of the GNSS station (Karegar et al., 2020). Additionally, we showed that identifying significant discontinuities in the GNSS timeseries due to antenna changes is not a straightforward task, leading to a relatively wide range of possible VLM rates between -0.18 $\mathrm{mm\,yr^{-1}}$ and 1.15 $\mathrm{mm\,yr^{-1}}$ (section 2.3). The picture of all VLM processes
ongoing at Terschelling could be further improved by including InSAR (Interferometric SAR) data and GIA (Glacial Isostatic Adjustment) models.

The computation of shorelines as the intersection between land elevation data and a horizontal plane at sea level height ("Jarkus shoreline") was limited to the JARKUS transects with spacings of about 250 m. This could potentially be improved
by using a gridded digital elevation model, if available in the required horizontal resolution and vertical accuracy, and applying image classification methods as was done for example by Liu et al. (2007) and Yousef et al. (2013). Additionally, the computation using the function from the JAT toolbox is limited by the JARKUS cross-shore resolution of 5 m, therefore the uncertainty for a single shoreline position can be up to $\pm$ 2.5 m. This could be improved by implementing a linear regression technique as presented by Stockdon et al. (2002).


Due to the complex morphology at the eastern and western tip of Terschelling (see pictures in figure A7), deriving satellite-derived shorelines from Landsat turned out to be a challenging task. As a result, the cross-shore timeseries based on shorelines from CASSIE in these areas exhibited discontinuities with magnitudes of several hundred meters. Ideas to improve cross-shore timeseries of satellite-derived shorelines comprise post-processing steps such as outlier removal, or experimenting with differ-
ent shoreline extraction algorithms, as well as using higher resolution optical sensors.

For all timeseries of cross-shore changes, we've subjectively selected a subset of transects used in the curved coastline sections of the eastern and western tip of the island. Therefore all given trends averaged over certain parts of the coastline might change with a different choice of transects. Another arbitrary processing decision whose influence we didn't investigate further
was the rejection of horizontal tidal corrections for satellite-derived shorelines that exceed $\pm$ 100 m.

**Transferability to other sites**

The coast of Terschelling offers contrasting conditions, such as retreating and advancing areas, or a straight central coastline
and more complex configurations especially at the Western tip of the island. However, the impact of sea level rise on the shoreline position depends on a variety of local factors, such as the type of sediment and the volume of the available sediment budget, the shape, orientation and exposure of the coastline, the hydrodynamic conditions such as tidal range, relative sea level changes, wave energy, currents and possibly also climate modes such as the NAO, the presence of rivers, vegetation or morphological features like dunes or sandbars, episodic extreme events like storm surges, and finally human impacts (e.g. Toimil et al., 2020;
Ranasinghe, 2016; Le Cozannet et al., 2014; Almar et al., 2023; Vousdoukas et al., 2023). Our conclusions for Terschelling that morphodynamics were responsible for the larger part of the shoreline changes between 1992 and 2022 can therefore not be

transferred to other study sites and other time periods. Nevertheless, the methodology to determine the geometrical influence of sea level change and morphodynamics using land elevation data, altimetry and satellite-derived shorelines can in principal be applied to all sandy coasts, under the condition that the observed shoreline and sea level changes exceed the uncertainty ranges.


The main limitation to transferability is the availability of land elevation data in high spatial and temporal resolution with high accuracy. While such data are available locally (e.g., Aquitaine in France (Nicolae Lerma et al., 2022), Narrabeen beach in Australia (Turner et al., 2016), Duck in USA (Larson and Kraus, 1994)), global datasets that cover also countries with less financial means are scarce. An alternative to land elevation data from in-situ and airborne LiDAR observations could be to estimate the topobathymetry from satellite remote sensing (e.g. Salameh et al., 2019; Gao, 2009). The topography can for example be derived from altimetry (e.g. Salameh et al., 2018), InSAR (e.g. Choi and Kim (2018)), stereo imagery (e.g. Almeida et al., 2019) or from a combination of sources (e.g. Pronk et al., 2024). For the bathymetry, there are different techniques that exploit the reflectance values from optical satellite imagery (e.g. Stumpf et al., 2003), that identify wave characteristics in optical or in SAR images (e.g. Bergsma et al., 2019), or that use a combination of radiometry and wave kinematics (e.g. Najar et al., 2022). For intertidal zones, different studies exploited the corresponding tidal variability of shorelines and sea level (e.g. Bishop-Taylor et al., 2019b; Chen et al., 2023), for example by assigning sea surface heights to instantaneous shorelines ("waterline method", e.g. Mason et al. (1995)).

To conclude, our findings quantify the geometric interplay between coastal inundation by sea level changes and morphological processes at Terschelling over the last three decades. The data-rich Dutch coast proved to be a valuable case study in that we were able to illustrate uncertainties in the remote sensing data sets compared to the available in-situ and LiDAR data. This paper provides a starting point to study the influence of sea level changes and morphodynamics in other regions, including those which have less local datasets.

*Code and data availability.* The code used to produce the results of this paper is written entirely with open source python packages and can be found in the github repository https://github.com/3enedix/P1-data-combination-code.git. Data produced in this paper can be found in a 4TU.ResearchData repository https://data.4tu.nl/, to be created during publication.

**Appendix A**

## A1 Cleaning GNSS height discontinuities

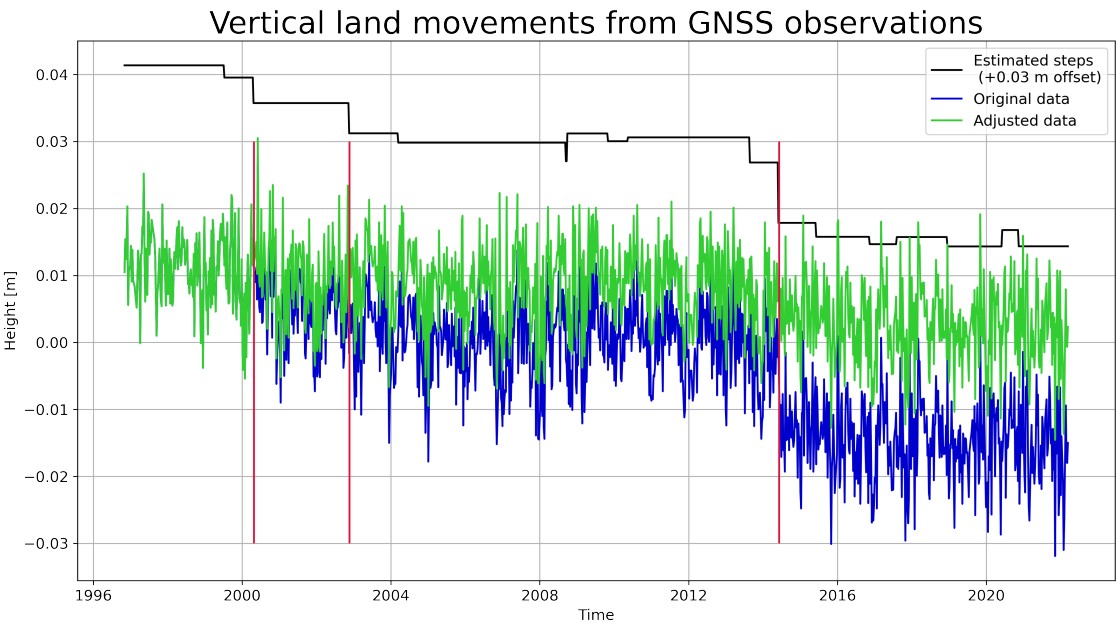

**Figure A1.** Vertical land motion from GNSS (NGL solution) at Terschelling. We estimate a step function with offsets for every indicated date of antenna or receiver changes. Most of these jumps are not significant. We therefore subsequently remove the three biggest offsets indicated by the vertical red lines. The green curve shown here is the result when removing all three offsets, resulting in a vertical land motion trend of -0.40 $\mathrm{mm\,yr^{-1}}$.

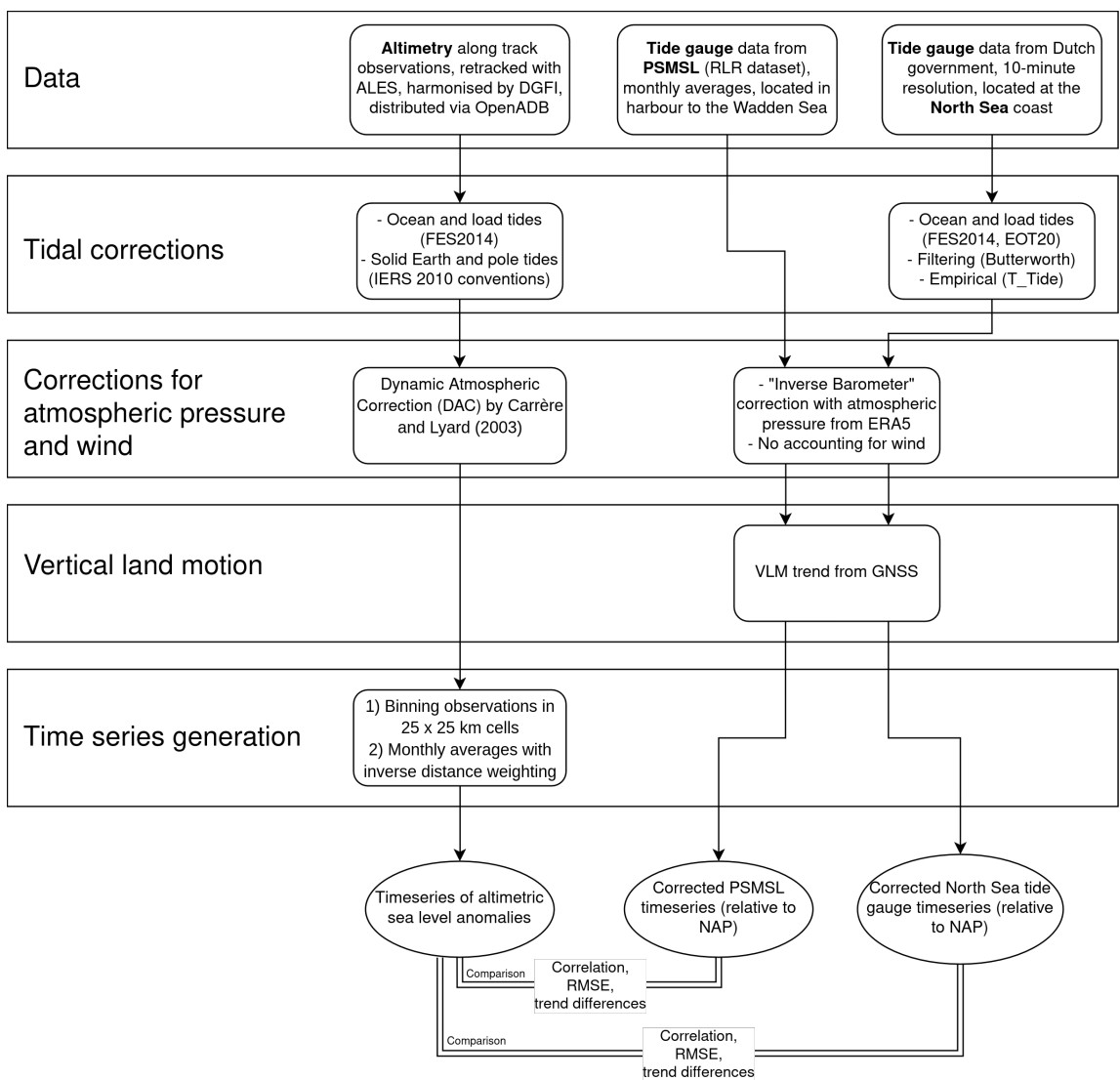

**Figure A2.** Workflow to make observations of sea level change from altimetry and from tide gauges comparable. See sections 2.1, 2.2, 2.3 and 3.1.

| cell | R | cell | RMSE [m] | cell | trend difference [mm yr$^{-1}$] |
|---|---|---|---|---|---|
| 95 | 0.70 | 95 | 0.09 | 52 | -1.1 |
| 83 | 0.67 | 83 | 0.09 | 70 | -1.0 |
| 96 | 0.67 | 96 | 0.10 | 14 | -1.9 |
| 84 | 0.64 | 55 | 0.10 | 78 | -2.6 |
| 55 | 0.62 | 82 | 0.10 | 3 | -1.2 |
| 82 | 0.62 | 67 | 0.10 | 77 | -1.1 |
| 67 | 0.62 | 4 | 0.10 | 15 | -1.3 |
| 78 | 0.61 | 69 | 0.10 | 44 | -1.4 |
| 69 | 0.61 | 78 | 0.10 | 95 | -0.6 |
| 15 | 0.59 | 56 | 0.10 | 69 | -1.7 |

**Table A1.** Linear correlation coefficient R, RMSE and difference in linear trend between altimetry timeseries and the PSMSL tide gauge for the 10 best scoring cells, respectively.

| cell | R | RMSE [m] | trend difference [mm yr$^{-1}$] |
|---|---|---|---|
| 96 | 0.67 | 0.10 | -1.9 |
| 95 | 0.70 | 0.09 | -1.1 |
| 83 | 0.67 | 0.09 | -1.0 |
| 82 | 0.62 | 0.10 | -1.1 |
| 70 | 0.66 | 0.10 | -0.2 |
| 69 | 0.68 | 0.09 | -0.6 |

**Table A2.** Linear correlation coefficient R, RMSE and difference in linear trend to the PSMSL tide gauge for the six cells forming the North East region.

| cell | R | RMSE [m] | trend difference [mm yr$^{-1}$] |
|---|---|---|---|
| 55 | 0.62 | 0.10 | -1.2 |
| 67 | 0.62 | 0.10 | -1.4 |
| 78 | 0.61 | 0.10 | -1.4 |

**Table A3.** Linear correlation coefficient R, RMSE and difference in linear trend to the PSMSL tide gauge for the three cells forming the middle region.

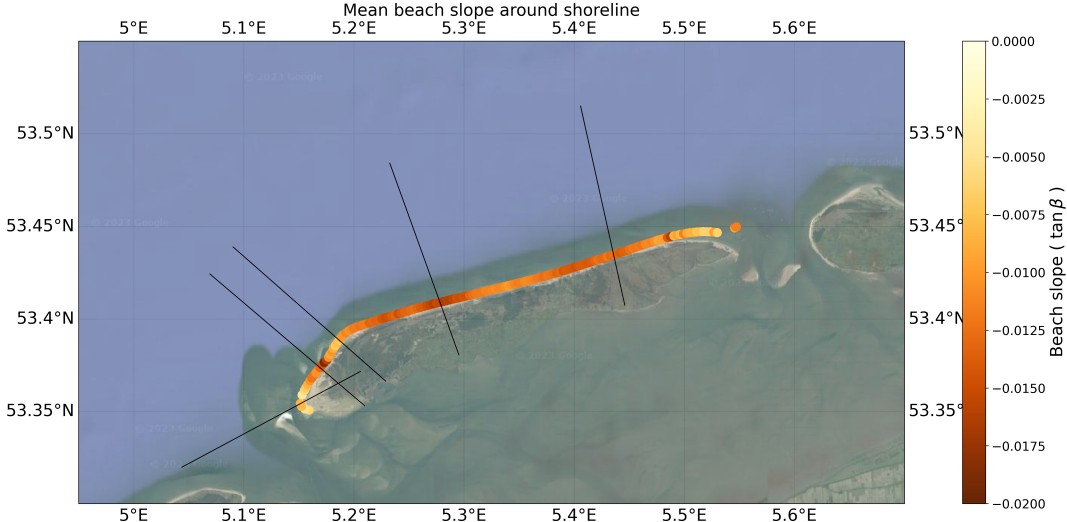

**Figure A3.** Mean beach slope per transect from the JARKUS data set. Beach slopes are especially mild at the outer parts that correlate with landward trends, and steeper to the middle of the coastline where the shoreline shows more seaward trends. The beach slopes in this figure are averaged in the area between the minimum and maximum shoreline position over the period 1992–2022. Background image from Google Map tiles using cartopy.io.img_tiles (© Google Maps)

## A3 Beach slope in JARKUS

## A4 Sensitivity analysis of cross-shore changes from CASSIE

## A5 Influence of waves on satellite-derived shorelines

The instantaneous shoreline positions extracted from satellite images are not only affected by tides, but also by high-frequency variations due to waves. To test the influence of waves on the cross-shore timeseries of satellite-derived shorelines from CASSIE, we computed the horizontal shift due to wave run-up and wave set-up.

For the computation of wave run-up we follow the empirical formula presented by Stockdon et al. (2006):

$$R_2 = 0.043\sqrt{HL}, \xi < 0.3$$

$$R_2 = 1.1(\eta_u + 0.5\sqrt{(0.75H\xi)^2 + (0.06\sqrt{HL})^2}), \xi \geq 0.3,$$

(A1)

where $R_2$ is the extreme run-up exceeded by 2% of the waves, $\xi$ is the surf similarity parameter or Iribarren Number $\xi = \frac{\beta}{\sqrt{H/L}}$ dependent on the beach slope $\beta$, $H$ is the significant wave height, $L$ is the wave length that can be computed as $L = \frac{gT^2}{2\pi}$ using the peak wave period $T$ and the gravitational constant $g$, and $\eta_u$ is the wave set-up derived here as

$$\eta_u = CH\xi,$$

(A2)

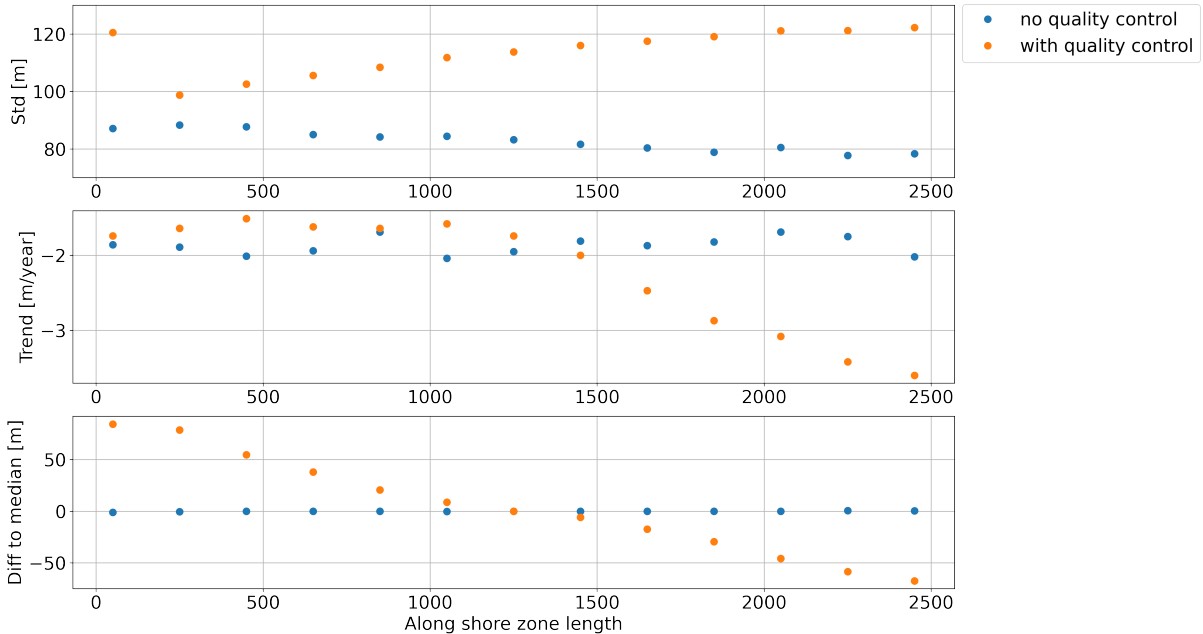

**Figure A4.** Sensitivity of satellite-derived cross-shore changes from CASSIE to changes in the along-shore zone length, the length of the zone used for the computation of one intersection between shoreline and transect. Presented are averaged statistics (standard deviation, trend and the difference to the mean of all solutions) for each solution computed as the median over all transects. An along-shore zone length between about 1000 m and 1500 m seems to be a reasonable choice with a small difference to median and stable trend estimates.

with $C$ being a constant between 0.15 and 4.


For the significant wave height $H$ and peak wave period $T$ we used ERA5 hourly data (variables "Significant height of combined wind waves and swell" and "Peak wave period") (Hersbach et al., 2022), interpolated to the time of image acquisition. For the beach slope, we used the time- and space variable beach slope derived from Jarkus in the 45 m buffer zone around the shoreline position used earlier for the tidal correction (Sec. 3.4 and 4.3). Similar to the tidal correction, we applied an arbitrary

threshold of $\pm50$ m in order to prevent extreme values due to small beach slopes and therefore division by almost infinitely small numbers (see Sect. 3.3).

The results for the full wave run-up corrections show that the horizontal shift for all transect-wise cross-shore timeseries has a median value of -15.0 m. When applying the wave run-up correction additionally to the tidal correction, the median of the

standard deviation of all transects increases from 82.2 m to 85.5 m, while the trends show no significant change at all.

Wave run-up is the maximum water level that is reached only for very short periods of time that are not necessarily the time of image acquisition. We therefore also tested the possibility of only correcting for wave set-up, the change in mean sea level

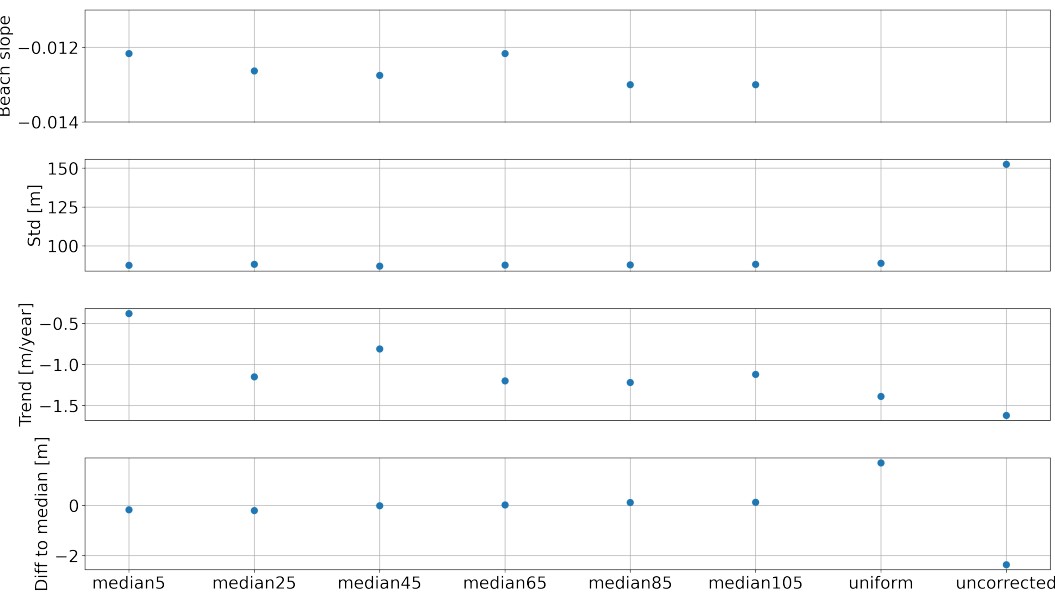

**Figure A5.** Sensitivity of satellite-derived cross-shore changes from CASSIE to different methods of tidal correction compared to using no tidal correction. 'median 5': Beach slope is computed as the median of all beach slopes in a buffer zone of $\pm$ 5 m around the shoreline position. We tested six buffer zone lengths between 5 m and 105 m. 'Uniform' : One beach slope ($tan\beta$ = - 0.01) for each transect and for each year. 'Uncorrected': No tidal correction. Statistics (standard deviation, trend and the difference to the mean of all solutions) are given as the respective median over all transects. The shown beach slope is computed as the median over time and the median per transect.

due to waves. The median horizontal shift due to wave set-up is 2.0 m, with the full range of values being between 0.6 m and 6.9 m. Considering the Landsat pixel size of 30 m and the best scoring RMSE in Almeida et al. (2021) of 8.84 m, it is not likely that correcting for wave set-up will improve the results. We see that the median standard deviation and trend do not change significantly (the standard deviation increases by 5 cm).

As the corrections for wave run-up or wave set-up slightly increase the noise in the timeseries or have no visible effect at all, we conclude that they cannot improve the cross-shore timeseries of satellite-derived shorelines from CASSIE.

## A6 Study site impressions

*Author contributions.* BA led the conceptualization and methodology development of the study, conducted investigations and formal analysis, wrote the required software, validated and visualized the results and wrote the original draft. RR and DvdW aided in the conceptualization and methodology development, and supervised the research activities. All authors edited and reviewed the manuscript. No artificial intelligence tools have been used to help with the creation of text or code.

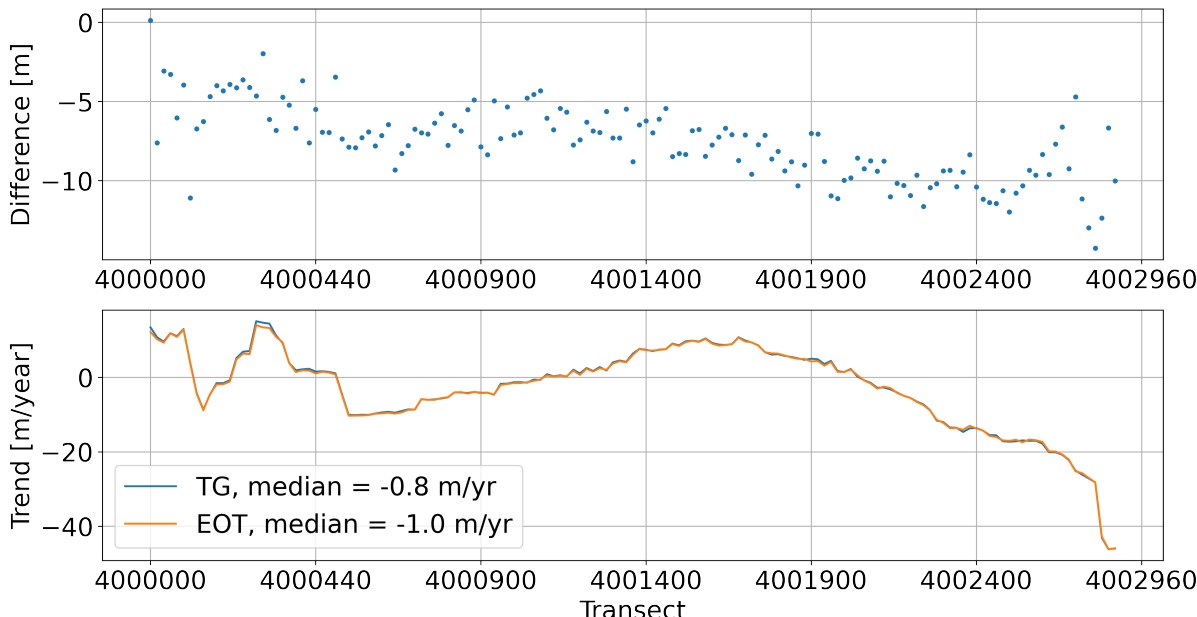

**Figure A6.** Sensitivity of tidally corrected cross-shore changes to the source of water level used for tidal correction. Upper: Median of absolute differences per transect between the cross-shore timeseries tidally corrected with water levels from tide gauge observations minus the version corrected using water levels from the EOT20 tidal model. Lower: Standard deviations of cross-shore timeseries tidally corrected with tide gauge data or EOT20.

*Competing interests.* The authors declare that they have no conflict of interest.

*Acknowledgements.* This work made use of freely available data and software. The ALES SSH data were produced by DGFI-TUM and distributed via OpenADB (http://www.openadb.dgfi.tum.de). More details on the retracker and the product are available in Passaro et al. (2014), Passaro et al. (2015) and Passaro (2017). The data from Hersbach et al. (2022) used to compute the inverted barometer correction contain modified Copernicus Climate Change Service information 2020. The code used to compute FES2014 was developed in collaboration between Legos, Noveltis, CLS Space Oceanography Division and CNES, and is available under GNU General Public License (https://www.aviso.altimetry.fr/en/data/products/auxiliary-products/global-tide-fes/description-fes2014.html). Furthermore, we used the python packages CoastSat (Vos et al., 2019b), Jarkus Analysis Toolbox (JAT) (van IJzendoorn, 2022) and matplotlib (https://doi.org/10.1109/MCSE.2007.55).


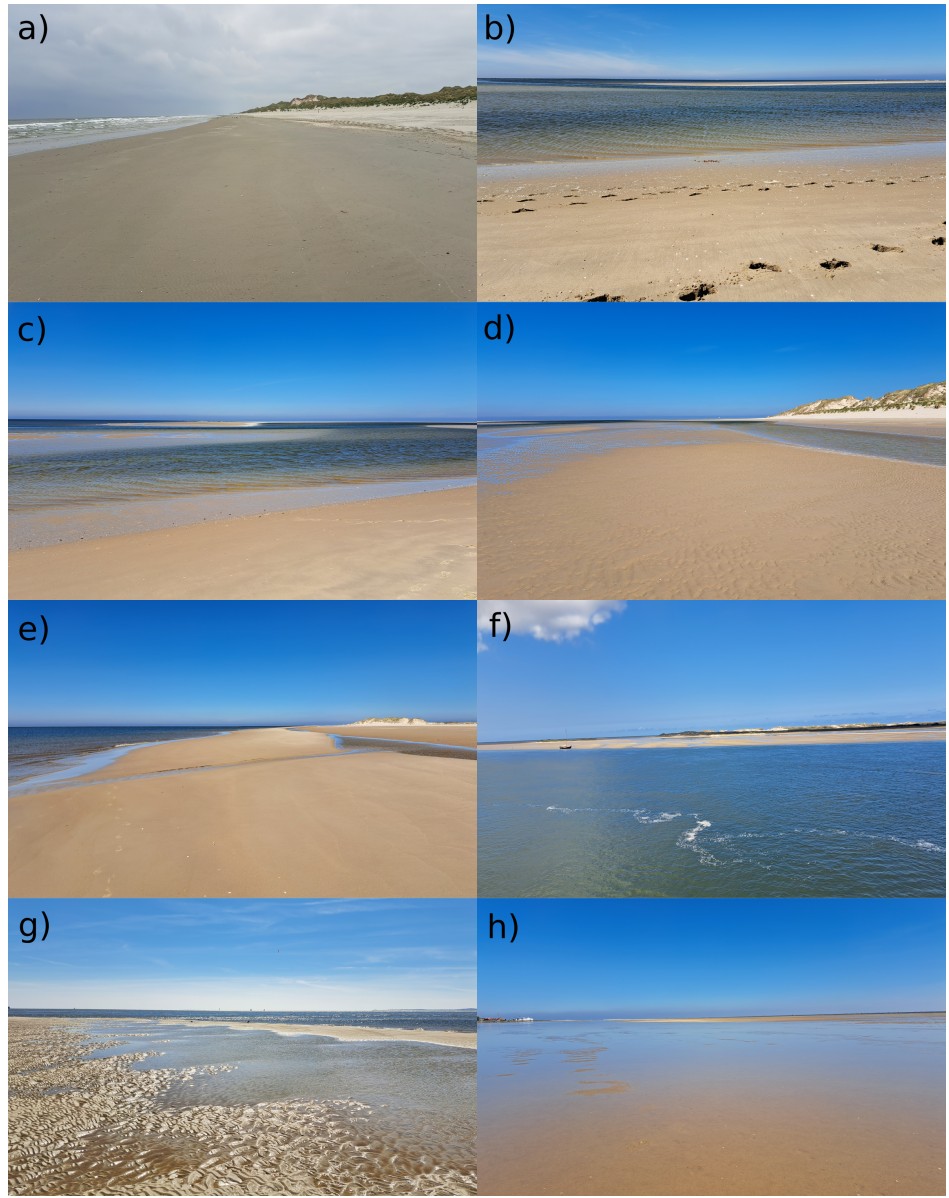

**Figure A7.** Impressions of the study site. a) Example cross-shore view from the central part of the Terschelling beach with the dunes to the right and the sea to the left. b) and c) View from the shore at the sandbanks between the central and the Western part of the beach. d) Cross-shore view when standing on a sandbank, with dunes, beach and water to the right, and patches of water and sandbanks in the distance. e) Cross-shore view of the shoreline close to the Western tip, with the sea to the left, patches of water in the beach to the right and a channel of water connecting them to the sea. f) View from the Wadden Sea onto the ebb tidal delta at the Vlie inlet to the West of Terschelling with a larger shoal (the Engelschoek) to the left. g) View from the beach to the Western tip with water and sand surfaces merrily mixing up, making the differentiation between "water" and "sand" almost impossible. h) View from the Western tip along the "Green Beach" ("Groene Strand") that is covered by patches of water and separates the dunes from the Wadden Sea.

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
