# Peer review of "Changing Sea Level, Changing Shorelines: Integration of Remote Sensing Observations at the Terschelling Barrier Island"

_EGUsphere, 2023_

## Author Comment (AC1)

**Response to reviewer 1, review 1 from 03.12.2023**

We would like to thank the reviewer for their helpful recommendations. We very much appreciate the time and effort they took to carefully read the manuscript and to formulate detailed and constructive suggestions on how to improve it.

In the following, you find our responses to the individual recommendations (in bold).

**0. The paper is in the direction of a community paper just published that they should cite in their manuscript: Laignel, B., Vignudelli, S., Almar, R., Becker, M., Bentamy, A., Benveniste, J., ... & Verpoorter, C. (2023). Observation of the coastal areas, estuaries and deltas from space. Surveys in Geophysics, 1-48**

Response
The suggested reference provides a helpful and relevant summary of observations of the coastal zone, especially remote sensing observations. We therefore added it in the introduction as a motivation for this study. The passage reads now as follows (with old parts in grey):

*Nowadays, there are several decades of remote sensing data available for coastal monitoring Laignel et al. (2023). Here we suggest to use observations for sea level and vertical land motion in combination with estimates of shoreline changes to quantify the geometrical relation between sea level and shoreline changes.*

**1. The authors use satellite radar altimetry to measure sea level. We know that this technique needs some specialized processing to retrieve data near coast. The authors use a coastal product that implements a retracker called ALES during two decades since 2002. However, the investigation of climate-related signals (e.g., trends) requires careful attention in ensuring homogeneity of processing between the various mission, removal of possible drifting in corrections, etc. For this reason ESA launched the Sea Level Climate change initiative to produce a validated gridded product that now is available through Copernicus (https://cds.climate.copernicus.eu/portfolio/dataset/satellite-sea-level-global). Moreover virtual stations are also provided here https://climate.esa.int/en/projects/sea-level/data/ . The authors should use these products as benchmark to assess that their processing chain is consistent.**

Response
In this study we used the OpenADB ALES-retracked along track product. In this product, the homogeneity of observations between different missions is achieved with cross-calibration with a global multi-mission crossover analysis (MMXO) (see Oelsmann et al. (2021), section 2.1). According to Oelsmann et al. (2021), possible drifts between the missions are almost completely removed by applying a radial correction to each single observation. A summary of the validation of this product is also provided in these slides: https://mediatum.ub.tum.de/doc/1446336/document.pdf

We considered using the ESA CCI virtual stations dataset at the beginning of the experiments, but it does not cover the study area.

We agree that a comparison of the along-track product with a gridded product, such as the one by ESA CCI might give further insights into uncertainties of coastal altimetry observations. However, as we found the area with the largest tide-gauge correlations is mainly positioned off-shore (> 50 km) on the open sea (see fig 4a), we don't expect too find very large differences between different altimetry products. We acknowledge that this could be different for other case studies. Nevertheless, in order to provide some guidance on this issue to the reader, we added the following paragraph in the new 'Limitations' section in the discussion:

*For deriving an altimetry timeseries, we restricted ourselves to the use of one single dataset. This is an along-track product retracked with ALES, an algorithm specifically designed for coastal areas, provided by the OpenADB (see section 2.1). This OpenADB ALES product has been used successfully before in studies combining altimetry and tide gauges (e.g. Mangini et al., 2022; Oelsmann et al., 2021). Our comparison to the local tide gauges and the use for computing the Jarkus shorelines in comparison to the other solutions showed that offshore altimetry can be used to study shoreline changes. However, in order to get the full picture of uncertainties in altimetry datasets, it could be useful to additionally include other products, such as the ESA CCI gridded product (Copernicus Climate Change Service, 2018).*

**2. The authors mentions various papers related to the synergy of altimetry, tide gauge and GNSS data. I suggest to integrate with recent papers that provide updated inverse methods aiming at a better characterization of the errors in estimating the sea level trends**

De Biasio F., Vignudelli S., Sea Level Change in the Mediterranean Sea from Satellite Altimetry and Tide Gauge. In Proceedings of Oceans from Space Conference (Editors: V. Barale, J.F.R. Gower, L. Alberotanza), 24-28 October 2022, Venice, Italy, 152-153, doi:10.57648/OceansFromSpaceV-2022-PROCEEDINGS.

De Biasio F., Vignudelli S., Baldin G.: Revisiting Vertical Land Motion and Sea Level Trends in the Northeastern Adriatic Sea Using Satellite Altimetry and Tide Gauge Data, Journal of Marine Science and Technology, 8(11), 949, doi:10.3390/jmse8110949, 2020.

Response
As both suggested references cover the same topic, we decided to include the latter one (De Biasio et al., 2020) in the introduction under point 1.1 Sea surface heights.

**3. The GNSS time series contains discontinuities from antenna and receiver changes. It should be recalled that GNSS is a point, sometime not co-located with tide gauge. Estimation of the VLM depends on how the station is managed and how logs a reupdated. We have seen differences between the various services around the world. Our feeling is that only local people can assess well the significance of VLM trends and errors. Sometime using InSAR can help, but I don't want ask authors to add these data if they are not expert with this technique. I just like authors inform readers about caveats when using GNSS stations. In Table 1, please add error to your VLM estimation (versions 1,2,3)**
Response

We agree that discontinuities in the GNSS timeseries and localised effects can be an issue when deriving rates of VLM, and that using only GNSS comes with some caveats. In this case study, the VLM rates are with magnitudes about 0.5 ± 0.4 mm/year relatively small compared to the rate of sea level rise with about 4.7 - 4.9 mm/year and therefore do not have a large effect on our main conclusions. To provide context for the reader, we added the following sentences to Section 2.3 Vertical land motion from GNSS (with old parts in grey):

*Here, we decide to manually remove one, two or three of the bigger offsets (with 9 mm, 4.5 mm and 3.8 mm respectively) in order to get a time series clean of artificial jumps but still containing the signal of VLM. The resulting VLM rates are summarised in Table 1, together with estimates from other publications for the same GNSS station. These estimates cover slightly different time periods, but when assuming that VLM rates are stable over approximately four years, we see a rather wide spread between -0.18 +/- 0.11 mm yr−1 (Gravelle et al., 2023, ULR7A) and -0.63 ± 0.43 mm yr−1 (Shirzaei et al., 2021). The differences in these outcomes of VLM rates indicate an uncertainty that approaches the magnitude of the signal. Another issue is that GNSS can only measure the component of VLM that takes place above the base of the GNSS station. Nevertheless, rates of GNSS height observations are currently the most accessible and up-to-date estimates of VLM. We therefore continue to work with the GNSS timeseries that results from removing the two largest offsets (version 2), as its VLM rate of -0.50 mm yr−1 fits best in the range of estimates from earlier publications.*

Furthermore, although we agree that the integration of more datasets could improve the understanding of all ongoing VLM processes at the study site we feel that it falls outside the scope of our paper. In order to provide some context on the shortcomings of using only GNSS as a source for VLM rates, we added the following paragraph in the new 'Limitations' section in the discussion:

*Another correction applied to the tide gauges for the comparison with altimetry was the vertical land motion (VLM). Here we used only data from a GNSS station as a proxy for VLM. However, this approach may neglect other ongoing processes such as sediment compaction below the base of the GNSS station (Karegar et al., 2020). Additionally, we showed that identifying significant discontinuities in the GNSS timeseries due to antenna changes is not a straightforward task, leading to a relatively wide range of possible VLM rates between -0.18  mm yr−1 and 1.15  mm yr−1 (section2.3). The picture of all VLM processes ongoing at Terschelling could be further improved by including InSAR (Interferometric SAR) data and GIA (Glacial Isostatic Adjustment) models.*

Regarding the error estimates, the standard deviations computed in the least squares estimate are with magnitudes of $10^{-4}$ unrealistically small as no error correlations are known and considered in the input data. We therefore computed the standard deviation based on the actual residuals, and are reporting these in the table.

**4. Detection of shoreline. Usage of state-of-the-art products is fine. However, Sentinel- 2 would provide more revisiting and better resolution. Landsat, like most other imaging satellites, is multispectral. The bands are Blue, Green, Red, near IR, and short wave IR, all with 30 m resolution There are one or two (depending on the**

**satellite) thermal IR bands with 60 m resolution There is a panchromatic image with 15 m resolution. In principle all spectral bands can contribute towards land-water discrimination, but in practice only a few bands provide robust and substantial leverage on classification land-water. Blue and green have the least contrast due to a combination of low and variable land albedo and possible strong and variable reflection from below water substrate. IR bands are generally better for water discrimination because there is a sharp increase in land albedo and increased absorption in water, leading to greater land-water contrast. There are two methods for improving the resolution: panchromatic sharpening and spectral un-mixing. The latest can improve detection from 30 m to 5 meters (see http://meetingorganizer.copernicus.org/EGU2013/EGU2013-9681.pdf ). I suggest authors to discuss a bit the various methods and highlights pros and cons about using a customized processing or using global products.**

Response
We appreciate the context provided by the reviewer and added some background and references for interested readers on optical satellite sensors and common methods to extract shorelines in the introduction subsection 1.2 Shoreline positions:

*Shoreline positions extracted from optical satellite images are in the preceding literature usually referred to as satellite-derived shorelines. When working with images from optical satellite missions, there is usually a trade-off between spatial resolution and revisit period. The group of sensors with moderate resolution (about 250 m - 1000 m pixel size), such as Terra/Aqua MODIS, Envisat MERIS or Sentinel3 OLCI, have high revisit periods (about 0.5 - 3 days), but images are usually too coarse to extract shoreline geometries with sufficient accuracy relative to the width of the beach. The most commonly used optical sensors for shoreline extraction are high resolution sensors (ca 5 to 30 m pixel size). Since 1999, these satellites often carry additional panchromatic sensors that generate black and white images with a finer resolution, and can be used to downscale the multispectral images. Examples are the long-term Landsat missions (30 m resolution of multispectral channels, with a 15 m panchromatic band) with a revisit period of 16 days, Sentinel-2 MSI  (10-20 m resolution) with a revisit period of 10 days (single satellite) or 5 days (two satellites in tandem) and long-term SPOT (5-20 m) with a revisit period of 26 days. Of these missions, SPOT is the only one whose data is not freely available. Finally, there are several commercial satellite missions with very high resolution (< 5 m) and short revisit periods (about 1-5 days) such as IKONOS, QuickBird, WorldView, or the cube satellite constellations by PlanetScope/Maxar. A more detailed review of optical satellite missions is given in Huang et al. (2018).*

*The process of extracting the shoreline from optical images starts usually by separating between land and water pixels. The easiest way to achieve this is to use a single band, which would preferably be one of the infrared bands where the differences in reflectance between water and land are the highest. The image of this band can be converted into a binary image by applying a threshold (e.g. Frazier and Page, 2000; Pardo-Pascual et al., 2012). This threshold can be chosen by a try-and-error procedure, or by computing it for example by using Otsu's method. Next to thresholding, the use of water indices (the ratios*

*of differences between bands) is very common to separate between land and water surfaces. There are several indices in use, where the choice depends on the type of the surroundings. For example, the Modified Normalised Difference Water Index (MNDWI) (Xu, 2006) was developed with the intent to better distinguish between water and buildings than the Normalised Difference Water Index (NDWI) (McFeeters, 1996). Usually the computation of an index is followed by the application of a threshold (e.g. Luijendijk et al., 2018; Dai et al., 2019; Almeida et al., 2021; Palomar-Vázquez et al., 2023), possibly also in combination with a classification (e.g. Vos et al., 2019b). Finally, there are advanced procedures to extract the shoreline at sub-pixel resolution, for example by using a marching squares algorithm to derive the shoreline contour (e.g. Bishop-Taylor et al., 2019a; Vos et al., 2019b) or by modelling the gradient of reflectances with polynomials and extracting the coordinates with the maximum gradient (e.g. Pardo-Pascual et al., 2012; Almonacid-Caballer et al., 2016; Sánchez-García et al., 2020).*

Regarding the choice of Landsat or Sentinel-2, as pointed out in the manuscript, Almeida et al. (2021) state in their paper describing CASSIE that they use surface reflectances for Landsat, but Top-of-Atmosphere reflectances when using Sentinel-2. As explained in the manuscript, we were not sure which consequences we can expect when mixing up results from surface and TOA reflectances in one timeseries, and therefore decided to use only one of the sensors. Sentinel-2 has higher temporal and spatial resolution, but Landsat provides the longer timeseries that is required when studying long-term changes in response to climate change.

**5. Comparison of altimetry with TG to estimate accuracy. I don't understand well how the tow measuring systems are made homogeneous. Comparison should be instantaneous. DAC and tides (if relevant) need to be removed as the two systems do not measure the same place. Some earth tides are seen partially by the TG. The recipe needs to be reported in appendix of the paper**

Response
Observations from altimetry and from the two tide gauges were made comparable in terms of signal content by applying corrections for tides, atmospheric pressure and vertical land motion as described in the data section. We've additionally added a flowchart in the appendix (figure A2) to clarify the procedure.

**6. Table 3 *[should probably be table 4]* : errors in trends need to be provided. Also significance of the trend should be checked (e.g. using the Mann–Kendall test).**

Response
We added the error margins for the trends of cross-shore changes derived from the intersection of JARKUS profiles with a plane at sea level. Similar to the error margins for vertical land motion computed under point 3), the standard deviations derived from error propagation were unrealistically small as no error correlations are known and considered in the input data, so we computed the standard deviation based on the distance of a single cross-shore estimate to the linear model. We're very thankful for the reviewer to point this out, as it led to the discovery of one single transect that caused error margins for the trend of 322 m/year. When looking at the timeseries in detail, we found that it exhibited unrealistic jumps over 2000 m. As the problem is confined to this single transect and we

cannot identify the cause for these jumps, we decided to exclude this transect from all further computations. We've added the following sentence to section 3.2 Cross-shore changes from the intersection of land elevation data (JARKUS) with sea level to justify the exclusion of this transect.

*Additionally, we also excluded one transect that exhibited unrealistic jumps larger than 2000 m from all computations.*

The exclusion of the faulty transect led to a small change in the numbers for the trends averaged over the entire coastline up to 0.5 m/year, where the largest change was in the results for intersections with altimetry and the tide gauge timeseries reduced to the altimetry time period. Consequentially, some of the statistics when comparing cross-shore timeseries from JARKUS and from CASSIE change as well. In detail, the average bias increased from -80.6 m to -82.8 m, and the trend differences increased from -2.1 m/year to -2.3 m/year. On the other side, their standard deviations decreased by 13 % and 2 %, respectively.

Regarding the significance of the trends, we added the averaged results of the Mann-Kendall test (with 1 meaning the trend is significant within the  5-95 % confidence interval and 0 meaning that no significant trend was detected) as suggested by the reviewer to table 4.

**7. A key point I would like mentioning is the replication of the approach to other sites and hopefully globally, following the promising validation in the study site. It is important to understand if a full remote sensing global application is feasible and if not the authors should explain how to fill the gaps. The authors highlight the need of land elevation data in high spatial and temporal resolution with high accuracy. Can SAR Interferometry fille the gap to measure land changes ? more and more SAR small satellites are going to be launched.**

Response
We agree that the lack of global land elevation data with high horizontal resolution and high vertical accuracy is a key limitation when transferring the methodology to other sites. We included a short summary of currently known approaches to estimate the different parts of the topobathymetry from satellite remote sensing in the discussion under "Transferability to other sites" (old parts in grey).

*The main limitation to transferability is the availability of land elevation data in high spatial and temporal resolution with high accuracy. While such data are available locally (e.g., Aquitaine in France (Nicolae Lerma et al., 2022), Narrabeen beach in Australia (Turner et al., 2016), Duck in USA (Larson and Kraus, 1994)), global datasets that cover also countries with less financial means are scarce. An alternative to land elevation data from in-situ and airborne LiDAR observations could be to estimate the topobathymetry from satellite remote sensing (e.g. Salameh et al., 2019; Gao, 2009). The topography can for example be derived from altimetry (e.g. Salameh et al., 2018), InSAR (e.g. Choi and Kim (2018)), stereo imagery (e.g. Almeida et al., 2019) or from a combination of sources (e.g. Pronk et al., 2024). For the bathymetry, there are different techniques that exploit the reflectance values from optical satellite imagery (e.g. Stumpf et al., 2003), that identify wave characteristics in optical or in SAR images (e.g. Bergsma et al., 2019), or that use a*

*combination of radiometry and wave kinematics (e.g. Najar et al., 2022). For intertidal zones, different studies exploited the corresponding tidal variability of shorelines and sea level (e.g. Bishop-Taylor et al., 2019b; Chen et al., 2023), for example by assigning sea surface heights to instantaneous shorelines ("waterline method", e.g. Mason et al. (1995)).*

**References**

Almeida, Luis Pedro, Israel Efraim de Oliveira, Rodrigo Lyra, Rudimar Luís Scaranto Dazzi, Vinícius Gabriel Martins, and Antonio Henrique da Fontoura Klein. "Coastal Analyst System from Space Imagery Engine (CASSIE): Shoreline Management Module." Environmental Modelling & Software 140 (June 1, 2021): 105033. https://doi.org/10.1016/j.envsoft.2021.105033.

Carrère, Loren, and Florent Lyard. "Modeling the Barotropic Response of the Global Ocean to Atmospheric Wind and Pressure Forcing - Comparisons with Observations." Geophysical Research Letters 30, no. 6 (2003). https://doi.org/10.1029/2002GL016473.

Mangini, Fabio, Léon Chafik, Antonio Bonaduce, Laurent Bertino, and Jan Even Ø Nilsen. "Sea-Level Variability and Change along the Norwegian Coast between 2003 and 2018 from Satellite Altimetry, Tide Gauges, and Hydrography." Ocean Science 18, no. 2 (March 18, 2022): 331–59. https://doi.org/10.5194/os-18-331-2022.

Oelsmann, Julius, Marcello Passaro, Denise Dettmering, Christian Schwatke, Laura Sánchez, and Florian Seitz. "The Zone of Influence: Matching Sea Level Variability from Coastal Altimetry and Tide Gauges for Vertical Land Motion Estimation." Ocean Science 17, no. 1 (January 12, 2021): 35–57. https://doi.org/10.5194/os-17-35-2021.

---

## Author Comment (AC2)

**Response to reviewer 2, review 1 from 15.05.2024**

We would like to thank the reviewer for their helpful recommendations. We very much appreciate the time and effort they took to carefully read the manuscript and to formulate detailed and constructive suggestions on how to improve it.

In the following, you find our responses to the individual recommendations (in bold):

1. **The shoreline concept and definition appear to be used ambiguously, requiring further clarification and objectivity. For instance, the terms 'shoreline' and 'coastline' seems to be used interchangeably, and comparisons between morphology-based, imagery-based, or elevation-based shorelines are not directly applicable (see [https://doi.org/10.2112/03-0071.1] [https://doi.org/1016/j.earscirev.2016.01.002] for reviews). How does this ambiguity relate to the reported bias? Additionally, can it justify the differences between tools, as mentioned by the authors, where 'the presented CASSIE-derived shoreline changes are only reliable over longer time periods, with CASSIE-derived shorelines being on average 39.2 m further seaward than the shorelines from CoastSat'? It seems that some differences arise because the authors are not measuring the same indicators.**

Response
We agree that the interchangeable use of the terms "coastline", "shoreline" (and also "land-water interface") is an issue in the existing literature. Inspired by Boak and Turner (2005), we were using these terms with the following meanings:
- Coastline: A  stretch along the coast, including both land and water surfaces.
- Land-water interface: The dynamic boundary between land and water
- Shoreline: A proxy for the ideal, instantaneous land-water interface which can be an idealised line like in this study, but could also be a pattern of inundated and dry areas

CASSIE and CoastSat are two different tools to extract shorelines from optical images. While the data and the target parameter are the same in both tools (delineate the border between sand and water in Landsat images), the implemented algorithms are quite different. In short, CASSIE  uses Otsu thresholding of the NDWI histogram of the image to create a binary image, from which polygons are extracted. The intersection of these polygons with pre-defined transects is the resulting shoreline proxy. CoastSat on the other hand first classifies the images (the classes are 'sand', 'water', 'white-water' and 'other land features') with a neural network classifier, before computing the Otsu threshold from an MNDWI image, and applying a marching squares algorithm to derive the contour along the found threshold. The bias we found between satellite-derived shorelines from CASSIE and from CoastSat is therefore most likely to be rooted in the differences between the respective algorithms.

When comparing satellite-derived shorelines and shorelines defined as the intersection between land elevation and a horizontal plane at sea level, we are indeed comparing two

different proxies for the shoreline position. However, both realisations capture the same morphological phenomenon.

For clarification, we've added the following paragraph in the introduction under the sub-heading "Shoreline positions":

*The terms "coastline", "shoreline" and "land-water interface" are not used uniformly in the existing literature. Inspired by e.g. Boak and Turner (2005), we use these terms here as follows. A coastline describes the stretch along the coast, including both land and water surfaces. The land-water interface is the dynamic boundary between land and water. The shoreline is a proxy for the ideal, instantaneous land-water interface. In this study, we use two different techniques to observe shoreline positions and their temporal evolution. These are the detection of shorelines from optical satellite images, and the derivation of shorelines by intersecting land elevation data with a plane at sea surface height. Both realisations of the shoreline position refer to the same geological feature. Their comparability depends on the respective observation uncertainties, the careful handling of different reference systems and the application of tidal corrections.*

2. **What is the influence of wave-runup on satellite derived the results? For wave dominated coasts this factor should be accounted for.**

Response
We would like to thank the reviewer to point out that waves can have an effect on satellite-derived shorelines, which we have not considered before. We therefore did additional computations with ERA5 hourly data of significant wave height and peak wave period, interpolated to the time of image acquisition, following the formulas by Stockdon et al. (2006), and computed the horizontal shift for each transect and each point in time. We found a median horizontal correction due to wave run-up of 15 m, which can be considered significant. However, applying this correction to the cross-shore timeseries of satellite-derived shorelines from CASSIE slightly increased the median standard deviation from 82.2 m to 85.5 m. There is no significant change in trends (median difference is 0.3 m/year). As the correction for wave run-up increases the noise of the timeseries (instead of reducing it as expected), we conclude that accounting for it would not improve the results.

We think the reason for this increase in noise could be that wave run-up is the highest possible water level that exists only during very short periods of time that are not necessarily the times of image acquisition. We therefore also tried to correct only for wave set-up in order to account for the change in mean sea level close to the coast due to waves. The horizontal shift due to wave set-up ranges between 0.6 m and 6.9 m, with no detectable changes in standard deviation or trends. We conclude that the correction for wave set-up is too small to be able to improve the results.

We added a section to the appendix (section A5) explaining all used formulas for wave run-up and wave set-up as well as the resulting corrections and their impact on the cross-shore timeseries.

3. **Authors should explain and justify why "the horizontal shift can become unrealistically large, especially for small beach slopes."**

Response
We've added some numbers to the respective paragraph to illustrate the magnitudes that extremely small beach slopes and the resulting corrections can take due to the numerical instability (division by very small numbers) that we consider not realistic. We are aware that it is hard to define quantitatively what is realistic or not. The choice of an arbitrary threshold is admittedly not a very elegant solution but one that seems us appropriate when using approximations for the beach slope and the resulting horizontal shift.

*The horizontal shift Δx resulting from the approximation formula (2) using the local beach slope can reach the physical limits of the beach if the local beach slope becomes very small. Some of the calculated beach slopes get as small as $8 \cdot 10^{15}$ (tan β) or even 0, leading to corrections up to 3.8 km or even infinity.* We therefore apply an *arbitrary threshold of ± 100 m for the maximum tidal correction.*

4. **In the manuscript the study area is introduced without any prior justification. The choice of Terschelling should be justified.**

Response
Thank you for pointing this out, this part got lost in the editing process. We've re-added the following sentences to the study area description:

*As a study area we chose the barrier island of Terschelling, that lies in a row of barrier islands separating the North Sea from the Wadden Sea at the Northern Dutch and German coast (Fig. 1a). We selected this study area because of its suitability for validating our method; it houses two tide gauges and a GNSS station, is covered by yearly LiDAR and bathymetry observations and its orientation is not parallel to the ground tracks of the satellite altimeters. This configuration allows us compare the respective local and remote-sensing observations, and to include the influence of vertical land motion. Additionally, Terschelling has a sandy beach, the type of beach that most available tools to extract satellite-derived shorelines are tailored to.*

5. **Oceanographic setting: authors state the "Short-term sea level variations at Terschelling are dominated by diurnal tides with a tidal range of 1.2 m–2.8 m during neap tide and spring tide, respectively.", but information on storm surge magnitude is also relevant. Concerning wave climate, period and direction characteristics are also needed.**

Response
We've added more information on the oceanographic setting (with old parts in grey):

*Short-term sea level variations at Terschelling are dominated by diurnal tides with a tidal range of 1.2 m-2.8 m during neap tide and spring tide, respectively. The average wave height is 1.5 m* *with a mean period of 8 seconds coming from west to north-east direction.*

*During storms, the wave heights can increase to 5-6 m, with an increased period of 10-15 seconds (Quataert et al., 2020).*

6.  **In my view, the title could be enhanced to better align with the paper's content. The remote sensing observations are not compared but integrated instead. Therefore, I propose revising the title to something like "Changing Sea Level, Changing Shorelines: Integration of Remote Sensing Observations at the Terschelling Barrier Island" for improved clarity.**

Response
The paper contains several data comparisons (tide gauge vs altimetry, satellite-derived shorelines from CASSIE vs CoastSat, satellite-derived shorelines from CASSIE with different processing parameters, CASSIE-derived shorelines vs Jarkus shorelines) and one data combination (land elevation + sea level to compute "Jarkus shorelines" from the intersection). We agree that the term "integration" can better reflect the range of comparisons and combinations, and changed the title of the manuscript accordingly.

7.  **A clear identification of major bottlenecks in the approach is missing.**

Response
We've added the following sub-section 'Limitations' as part of the discussion.

[revised manuscript text omitted]

8. **If space- and time-variable beach slope is available then information on satellite derived shoreline is generally not needed. That information is mainly needed for validation purposes.**

Response

We agree that satellite-derived shorelines are in this case not required to determine the influence of sea level rise and morphodynamics on the shoreline position, but serve here only as validation. However, the main purpose of the paper is not to determine the impact of sea level rise on shoreline positions at Terschelling, but to prepare synthesis methods of remote sensing datasets that can be applied to other coastlines of the world, and to show

that the Bruun Rule is not the only way forward. As satellite-derived shorelines can play an important role in areas where highly accurate land elevation data in high spatial and temporal resolution is not available, we wanted to illustrate the processing chain from using tools like CASSIE or CoastSat to the final cross-shore timeseries and especially their uncertainties.

9. **A figure of a representative cross-shore profile would be very useful in order to allow the reader to perceive the main morphological features, such as the presence of bars, beach face and berm characteristics, the existence of a dune.**

Response
Figure 8 of the manuscript shows three example profiles from each of the three sections exhibiting landward and seaward movements. The main intention of this figure was to illustrate the effect of the tidal correction, but it shows also some of the morphological features at Terschelling. We've updated the figure with texts and colours to clarify the situation. Additionally, we've added a panel of pictures in the appendix to give an impression of the study site, and made references to this in the main text.

10. **Closure depth should also be reported.**

Response
We added an estimate for the closure depth from a previous study to the study area description (section 1.3):

*The long-term closure depth is reported to lie between 4 m and 10 m, with a tendency for smaller values in the west and larger values to the east of Terschelling (Marsh et al., 1999).*

11. **Concerning the response to sea level rise is not clear if authors used or not the Brunn Rule (as it mentioned in the introduction but no other reference is made).**

Response
The second paragraph of the introduction introduces the drivers of shoreline change and quantifying their individual contributions. As the Bruun Rule is widely used to quantify the effect of morphodynamics and sea level rise on the shoreline, it seemed appropriate to give a short overview of its shortcomings and its widespread use in the current literature. In the third paragraph however we explain that we approach this task by using observations instead of models.

We've modified the beginning of the third paragraph as follows (old parts in grey) to make clear that the Bruun Rule is not used here:

*Nowadays, there are several decades of remote sensing data available for coastal monitoring (Laignel et al., 2023). Instead of using the Bruun Rule, we suggest an*

*alternative approach using observations* for sea level and vertical land motion in combination with estimates of shoreline changes to quantify the geometrical relation between sea level and shoreline changes.

12. **Referring to Brunel and Sabatier (2009), the authors assert that protected 'pocket beaches' are more susceptible to inundation from sea-level rise compared to open beaches, which are more affected by increased wave energy. However, this assertion may depend on the time scales considered. In sandy beaches with equilibrium morphological profiles, the ratio between shoreline retreat and sea-level rise is substantially higher than in pocket and platform beaches. For further insights on this topic, authors are encouraged to consult relevant references.**

Response

We agree that the example of pocket beaches studies by Brunel and Sabatier (2009) does not represent the full range of coastal settings. We therefore replaced this example with a short summary of site-specific factors that can influence the results (with old parts in grey):

*The coast of Terschelling offers contrasting conditions, such as retreating and advancing areas, or a straight central coastline and more complex configurations especially at the Western tip of the island.* However, the impact of sea level rise on the shoreline position depends on a variety of local factors, such as the type of sediment and the volume of the available sediment budget, the shape, orientation and exposure of the coastline, the hydrodynamic conditions such as tidal range, relative sea level changes, wave energy, currents and possibly also climate modes such as the NAO, the presence of rivers, vegetation or morphological features like dunes or sandbars, episodic extreme events like storm surges, and finally human impacts (e.g. Toimil et al., 2020; Ranasinghe, 2016; Le Cozannet et al., 2014; Almar et al., 2023; Vousdoukas et al., 2023). Our conclusions for Terschelling that morphodynamics were responsible for the larger part of the shoreline changes between 1992 and 2022 can therefore not be transferred to other study sites and other time periods. *Nevertheless, the methodology to determine the geometrical influence of sea level change and morphodynamics using land elevation data, altimetry and satellite-derived shorelines can in principal be applied to all sandy coasts, under the condition that the observed shoreline and sea level changes exceed the uncertainty ranges.*

Minor comments:

1. Line 4 – Knowing "about the individual contributions of sea level change, vertical land motion and morphodynamics " is not only essential to "necessary to make informed choices when applying coastal defence measure" but also for selecting other adaptation options.

Response
We've changed the respective sentence as follows (old parts in grey):

*Therefore, knowledge about the individual contributions of sea level change, vertical land motion and morphodynamics on shoreline changes is necessary to make informed choices for climate change adaptation, such as applying coastal defence measures.*

2. In figure 2, should "information" replace "output data "?

Response
In our understanding, "information" is a more generic category that entails sub-categories such as "output data". We would like to keep the more specific term "output data", also to make the distinction between input from other sources and output after applying the methods from this paper more clear.

3. The number of significative figures in "g" is exaggerated.

Response
Thank you for pointing this out. Indeed our inner geodesist might have gotten a little overexcited here. We reduced the number of digits after the comma to the common g = 9.81 m/s$^2$.

4. Long-shore should be replaced by longshore.

Response
We've replace the word "long-shore" with the word "longshore".

5. The phrase "the tide gauge observations have to be corrected for vertical land motion" does not apply if we are interested in relative sea level – which is the case when considering sea level influence on coastal change. This is because the focus is on the changes in sea level relative to the land surface rather than absolute sea level measurements.

Response
We agree that the study of shoreline changes requires relative sea level observations that are not corrected for VLM. These thoughts are partly reflected in the different solutions of Jarkus shorelines, where we used (among others) both, corrected and uncorrected tide gauge data in order to see there differences (which were small, see section 4.2). Furthermore, for the comparison with cross-shore changes from CASSIE, we use solution created with uncorrected tide gauge data (see section 3.5).

The cited sentence comes from the section "2.2 Sea level heights from tide gauges", which explains how we make tide gauge observations comparable to altimetry observations, in order to validate the latter. To avoid comparing relative sea level with absolute sea level, we bring these two together by correcting for vertical land motion.

6. Separating data from methods can complicate the text's coherence and flow.

Response
We generally agree that having the data description and the methods together in one section can help the reader's understanding. In this case we chose to separate data and methods, as we were using several different datasets that had already undergone some processing. We therefore decided to use the "data" section to describe pre-processing steps that we implemented but are not novel, as well as processing that was done by someone else, and use the "methods" section to describe our own processing.

We've added the following half-sentence to the introduction to point this structure out to the reader (old parts in grey):

*After describing the datasets and the required post-processing steps in section 2, we start by evaluating the ability of offshore altimetry observations to capture sea level variations at the coast by comparing altimetric sea level anomalies to sea surface heights from tide gauges (Sect. 3.1).*